# FORTE 2.0: a fast, parallel and flexible coupled climate model

Adam T. Blaker[1], Manoj Joshi[2], Bablu Sinha[1], David P. Stevens[3], Robin S. Smith[4], and Joël J.-M. Hirschi[1]

[1]National Oceanography Centre, European Way, Southampton, SO14 3ZH, UK
[2]Climatic Research Unit, School of Environmental Sciences, University of East Anglia, Norwich, UK
[3]Centre for Ocean and Atmospheric Sciences, School of Mathematics, University of East Anglia, Norwich, UK
[4]NCAS, Dept. of Meteorology, University of Reading, Reading, UK

*Correspondence to:* A. T. Blaker (atb299@noc.ac.uk)

**Abstract.**

FORTE 2.0 is an intermediate resolution coupled Ocean Atmosphere General Circulation Model (AOGCM) consisting of IGCM4, a T42 spectral atmosphere with 35 $\sigma$ layers, coupled to MOMA, a $2° \times 2°$ ocean with 15 z layer depth levels. Sea-ice is represented by a simple flux barrier. Both the atmosphere and ocean components are coded in FORTRAN. It is capable of producing a stable climate for long integrations without the need for flux adjustments. One flexibility afforded by the IGCM4 atmosphere is the ability to configure the atmosphere with either 35 $\sigma$ layers (troposphere and stratosphere) or 20 $\sigma$ layers (troposphere only). This enables experimental designs for exploring the roles of the troposphere and stratosphere, and the faster integration of the 20 $\sigma$ layer configuration enables longer duration studies on modest hardware. A description of FORTE 2.0 is given, followed by analysis of a 2000 year long control integration.

## 1 Introduction

Numerical models of the coupled (ocean-atmosphere) climate system are important tools for studying the Earth's climate. They provide insight into phenomena which are difficult to observe directly, such as the Atlantic Meridional Overturning Circulation (AMOC) or the effects of increasing atmospheric $CO_2$. They can also be used to test hypotheses about the global climate and the world we live in. In a climate model it is possible to study the climate response to extreme events such as loss of ice cover (and the resulting change in albedo). Whilst the results in terms of quantitative changes to temperature, precipitation, and other climate variables should be treated with caution, it is possible to examine the processes which lead to the predicted changes.

There is a broad spectrum of coupled climate models. At one end of the spectrum are coarse resolution simplified models designed to run millennial scale experiments quickly and for minimal computational cost, both in terms of computing power and memory resources (e. g. GENIE (Marsh et al., 2007), CLIMBER (Montoya et al., 2005), UVic (Weaver, 2004), ECBilt (Haarsma et al., 1996)). At the opposite end of the spectrum, high resolution ($<0.1°$- $0.5°$ ocean) models (e.g. those contributing to the CMIP6 HighResMIP (Haarsma et al., 2016) such as HadGEM3-GC3.1 (Roberts et al., 2019), and HiGEM (Shaffrey et al., 2009)). Between the two extremes are the intermediate resolution models (e.g. HadCM3 (Gordon et al., 2000), FAMOUS (Smith et al., 2008)), including most of the coupled climate models contributing to CMIP3 (Meehl et al., 2007), CMIP5 (Taylor

et al., 2012) and CMIP6 (Eyring et al., 2016). It is worth noting that model development does not equate solely to increase in horizontal resolution. Inclusion of more, or better parameterised, Earth System processes can be equally if not more important developments (e.g. Sellar et al., 2019).

Sinha and Smith (2002) developed FORTE (Fast Ocean Rapid Troposphere Experiment), a fast and flexible coupled climate model, for the purposes of climate studies. FORTE's speed and flexibility meant that the original model was an ideal educational and research tool. The flexibility of FORTE is evident in the variety of experiments in which it has been used to study ocean and/or climate phenomena. Examples include: Buchan et al. (2014), in which observed SST anomalies from 2009-10 were applied to the model SST field prior to the coupling time step, enabling the authors to examine the effect of observed SST anomalies in an otherwise free-running coupled model; Sinha et al. (2012), in which the simplicity of FORTE made it easy to examine the effect of orography on the Atlantic meridional overturning circulation; Wilson et al. (2009) used the flexibility to examine the roles of orography and ocean dynamics on atmospheric storm tracks, performing experiments with either an interactive ocean or the static mixed layer option in the IGCM atmosphere; Blaker et al. (2006) added initial condition perturbations to the Southern Ocean to study fast ocean teleconnection processes and Atkinson et al. (2009) performed similar experiments using both ocean-only and coupled configurations. Although most studies used FORTE with a T42 resolution atmosphere and 2° ocean, a version of FORTE using a T21 resolution atmosphere and 4°ocean has also been used for idealised experiments featuring Pangean and aquaplanet configurations (e.g. Smith et al., 2004, 2006). Simulations of FORTE have also been analysed in other climate studies (Hunt et al., 2013; Grist et al., 2008). However, until now there has been no comprehensive, peer-reviewed publication describing the model itself.

A new version of the atmosphere component of FORTE was released in 2015, and a desire to perform coupled experiments once again resulted in a refresh of FORTE. To avoid confusion with earlier endeavours, but at the same time make clear the ancestry of the model we decided to refer to the refreshed model as FORTE 2.0. This paper describes the coupled model and its components and demonstrates that FORTE 2.0 produces a realistic and stable climate without the need for flux adjustments. The control integration described is a 2000 year long integration starting from rest with the Levitus (temperature, salinity) climatology (Levitus and Boyer, 1998; Levitus et al., 1998) and pre-industrial atmospheric concentrations of $CO_2$. The model is forced solely by incoming solar radiation at the top of the atmosphere. The rest of this paper is organised as follows: Section 2 gives a description of FORTE 2.0; Section 3 presents the model spin-up; Section 4 presents the control climate; Section 5 discusses the main modes of climate variability in the model; Section 6 concludes.

## 2   Model Description

FORTE 2.0 is a global coupled ocean-atmosphere general circulation model consisting of a 2° resolution configuration of the MOMA (Modular Ocean Model - Array) (Webb, 1996) ocean model coupled to a T42 (approximately 2.8°) configuration of the IGCM4 (Intermediate General Circulation Model 4) (Joshi et al., 2015) atmosphere model. FORTE 2.0 is an updated incarnation of FORTE (Fast Ocean Rapid Troposphere Experiment) (Sinha and Smith, 2002; Smith et al., 2004), with the most significant change being an update of the atmosphere from IGCM3 (Forster et al., 2000) to IGCM4 (Joshi et al., 2015).

The ocean and atmosphere components of FORTE 2.0 are coupled once per model day using OASIS version 2.3 (Terray et al., 1999) and PVM version 3.4.6 (Parallel Virtual Machine, see http://www.csm.ornl.gov/pvm/, Geist et al. (1994)). Integration is relatively fast (~100 model years per wallclock day on a 28 core 2.4GHz Intel Broadwell CPU) and the model can be run on a desktop computer, making it ideal for experiments where more complex higher resolution models are resource limited.
The retention of the full primitive equations for fluid flow in both atmosphere and ocean allows more realistic simulations than possible with Earth Models of Intermediate Complexity (EMICs). In addition, FORTE 2.0 is readily configurable, allowing experiments with realistic and idealized configurations of coastlines, orography, and ocean bottom topography.

## 2.1 The atmosphere component

The atmosphere component of FORTE 2.0 is IGCM4 (Joshi et al., 2015), run with a T42 spectral resolution. A longitudinally
regular and Gaussian in latitude grid with a grid spacing of 2.8° is used for advection and diabatic processes. The resolution is sufficient to enable stable climate integrations without the need for flux adjustments. There are two pre-configured choices for the number of vertical levels: a troposphere only atmosphere represented by 20 $\sigma$ levels (L20) which extends to around 25 km altitude, or a 35 $\sigma$ level configuration (L35) which includes the stratosphere and extends to around 65 km altitude. To avoid issues with $2\Delta z$ oscillations under certain conditions the NIKOSRAD radiation scheme in IGCM3 (used previously in
FORTE) was replaced with a modified version of the Morcrette radiation scheme (Zhong and Haigh, 1995). For further details of the IGCM4 and its performance we refer the reader to Joshi et al. (2015) and references therein. The model is run with 96 (L35) or 72 (L20) time steps per day. Orography is derived from the US Naval 1/6[th] degree resolution dataset. IGCM4 is MPI parallelised, and at this resolution integration on 16-32 cores achieves the best performance.

Atmospheric convection is dealt with via a Betts-Miller scheme (Betts and Miller, 1993). Low, medium and high layer cloud
and convective clouds amount are represented, based on a critical relative humidity criterion (see the Appendix of Forster et al. (2000)). The formula which determines low-level cloud amount has an additional factor of 50% compared to that used by Forster et al. (2000) to correct a cold bias within the tropical ocean which led to unrealistic circulation in the Pacific. In addition to variation with solar zenith angle (and hence latitude), sea surface albedo is increased away from polar regions to compensate for the absence of aerosols which would otherwise scatter incoming solar radiation. Land grid boxes are assigned
a vegetation index, one of 24 pre-defined vegetation types, which determine the albedo and roughness length.

Coupling to the dynamic ocean model requires some changes to the surface boundary layer. In order to conserve water it is necessary to account for soil moisture and implement river runoff. Soil moisture for each land grid box is represented as a bucket, or reservoir, 0.5 m in depth. Excess water, i.e. when the volume of water is greater than the volume of the bucket, is accumulated and added to the ocean as runoff at each coupling timestep. The land surface is divided into catchment basins
and the accumulated runoff is distributed between a list of ocean cells that represent river mouths. Runoff accumulated over Antarctica is distributed uniformly over the ocean south of 55°S, as a simplistic representation of iceberg calving and melting. Additionally, land snow cover is capped at a maximum thickness of 4 m. Excess snow over Antarctica and the Arctic region is treated separately as an additional runoff term that represents iceberg melting and calving. As with the soil moisture, runoff

from excess snow over Antarctica is distributed uniformly over the ocean south of 55°S. Excess snow melt over the Arctic is handled similarly, with a uniform distribution over the ocean north of 66°N.

To improve the representation of the effects of sea surface roughness on momentum exchange a wind-dependent drag co-efficient, $C_d$, is implemented, such that $C_d = C_d^0 + 5.6e^{-5}*$windspeed (Wu, 1980). This gives a maximum $C_d = 0.003$ at windspeed of 40 m s$^{-1}$ without ice cover. $C_d^0$ is the drag coefficient over ocean cells, calculated using a globally uniform value for surface roughness over open ocean.

At present FORTE 2.0 does not include dynamic sea-ice representation. Instead, sea ice is represented by a barrier to heat fluxes between the ocean and atmosphere component, which is imposed when the sea surface temperature reaches 271 K, and surface albedo is increased to 0.6 to represent ice cover. The albedo continues to linearly increase, reaching 0.8 at 261 K as a means to represent the albedo effects of snow on ice. Once the albedo reaches 0.8 it will not reduce until the temperature rises above freezing point and the flux barrier deactivates. There is no advection of sea ice, and salinity and runoff fluxes remain unaffected. Sea-surface temperature (SST) under ice is relaxed toward the freezing point of seawater (-1.8° C) on a 10-day timescale.

## 2.2 The ocean component

The ocean component of FORTE 2.0 is MOMA (Modular Ocean Model - Array, Webb (1996)), a version of the GFDL MOM (Modular Ocean Model) (Pacanowski et al., 1990) coded to work more efficiently on array processors, which solves the primitive equations discretized using finite differences on an Arakawa B grid (Arakawa, 1966). It has a linear free surface (Killworth et al., 1991) and uses 'full cell' ocean bathymetry. In the configuration used for this integration the ocean horizontal resolution is 2° x 2°, with 15 $z$-layer levels, increasing in thickness with depth from 30 m at the surface to 800 m at the bottom. A polar island, comprising the top row of grid cells (88-90°N), is required in the Arctic to prevent numerical instability due to convergence of lines of longitude. There are 64 baroclinic time steps per day (22.5 minute time steps) implemented using the modified split QUICK (MSQ) advection scheme (Webb et al., 1998). MOMA is parallelised using OpenMP and running on 4-6 cores is typically sufficient to match the IGCM4 performance.

Bathymetry is derived from the ETOPO5 (1988) 1/12° resolution dataset, and interpolated onto the model resolution. Due to the horizontal resolution, in order to encourage dense water formation and flow between the Nordic Seas and North Atlantic, bathymetry is manually excavated in a manner similar to HadCM3 (Gordon et al., 2000). The Bering, Gibraltar and Katte-gat/Skagerrak Straits are represented by a single grid box which, due to the Arakawa B grid, means that there is no advection through them, but diffusion of potential temperature ($T$) and salinity ($S$) does occur.

Ocean isopycnal mixing is represented in MOMA through the isoneutral mixing scheme of Griffies et al. (1998). The eddy-stirring process of Gent and McWilliams (1990) is introduced as a skew flux (Griffies, 1998). Where isopycnal slopes become large, exponential tapering scales isoneutral diffusivities to zero as the slope increases (Danabasoglu and McWilliams, 1995). The isopycnal mixing parameters used for the control simulation described in section 3 are shown in Table 1.

To ameliorate some of the shortcomings identified in earlier FORTE simulations some additional changes have been made to MOMA. Firstly, the background vertical diffusion, $\kappa$, is set to be stability dependent (Gargett, 1984), albeit with the surface

| Parameter | Value ($m^2 \, s^{-1}$) |
|---|---|
| Horizontal viscosity | $4.0 \times 10^3$ |
| Isopycnal tracer diffusivity | $2.5 \times 10^3$ |
| Isopycnal thickness diffusivity | $2.0 \times 10^3$ |
| Steep slope horizontal diffusivity | $1.5 \times 10^3$ |
| Vertical viscosity coefficient | $1.0 \times 10^{-3}$ |
| Bottom drag coefficient | 0.001 |
| Max. slope of isopycnals | 0.002 |

**Table 1.** Mixing parameters in MOMA.

to sea floor potential temperature gradient as a simple proxy for stability, such that

$$\kappa_s = (0.3 + 1.7 \mathrm{e}^{-(0.15[\max(T_s, T_b) - T_b])^2}) \times 10^{-4} \, \mathrm{m^2 s^{-1}},$$
$$\kappa = \kappa_s + (2 \times 10^{-4} - \kappa_s) z / H \, \mathrm{m^2 s^{-1}},$$

where $T_s$ is the surface potential temperature, $T_b$ is the bottom potential temperature, $z$ is depth and $H$ is the local total depth

of the ocean. Thus the vertical diffusivity takes a maximum value of $2 \times 10^{-4} \, \mathrm{m^2 s^{-1}}$ at the sea floor and at high latitudes, with lower values, approaching $3 \times 10^{-5} \, \mathrm{m^2 s^{-1}}$ in the upper ocean at low latitudes.

Secondly, starting from 5° N/S the horizontal diffusion in the surface layer increases towards the Equator from its default value to 20 times this value at the Equator, to counteract equatorial upwelling and, in a simple way, parameterise the eddy heat convergence associated with tropical instability waves which was highlighted by Shaffrey et al. (2009).

## 3   Spinup of the control integration

To evaluate the performance of FORTE2.0 we run a pair of control integrations with pre-industrial $CO_2$ concentration using both the 35 $\sigma$ layer and 20 $\sigma$ layer atmosphere configurations. In each simulation FORTE 2.0 starts from rest with identical initial ocean temperature and salinity fields from Levitus and Boyer (1998) and Levitus et al. (1998) interpolated onto the ocean model grid. Figure 1 shows area and volume integrated quantities of the surface heat and fresh water fluxes, and ocean

temperature and salinity from each of the two integrations. The surface heat flux into the ocean is initially positive (up to 1.5 W m$^{-2}$) but the imbalance reduces to less than 0.5 W m$^{-2}$ after a few decades and then stabilises and remains within $\pm$ 0.2 W m$^{-2}$ throughout the remainder of the integration (Fig. 1 a)). The time average water budget closes to within -0.2 mm/year, after an initial adjustment in the first year of the integration (Fig. 1 b)). The global average sea surface temperature (SST) settles within 100 years to values around 19.1°C for the L35 configuration and 19.0°C for the L20 configuration (Fig. 1 c) .

Sea surface salinity (SSS) in both configurations adjusts more slowly, with L35 maintaining a value of around 35.15 PSU after 1000 years of integration and L20 approaching 35.23 PSU towards the end of the 2000 year simulation (Fig. 1 d)). The mean SST is 0.9°C warmer than the initial state provided by Levitus 1998 (18.2°C). The volume average ocean potential temperature

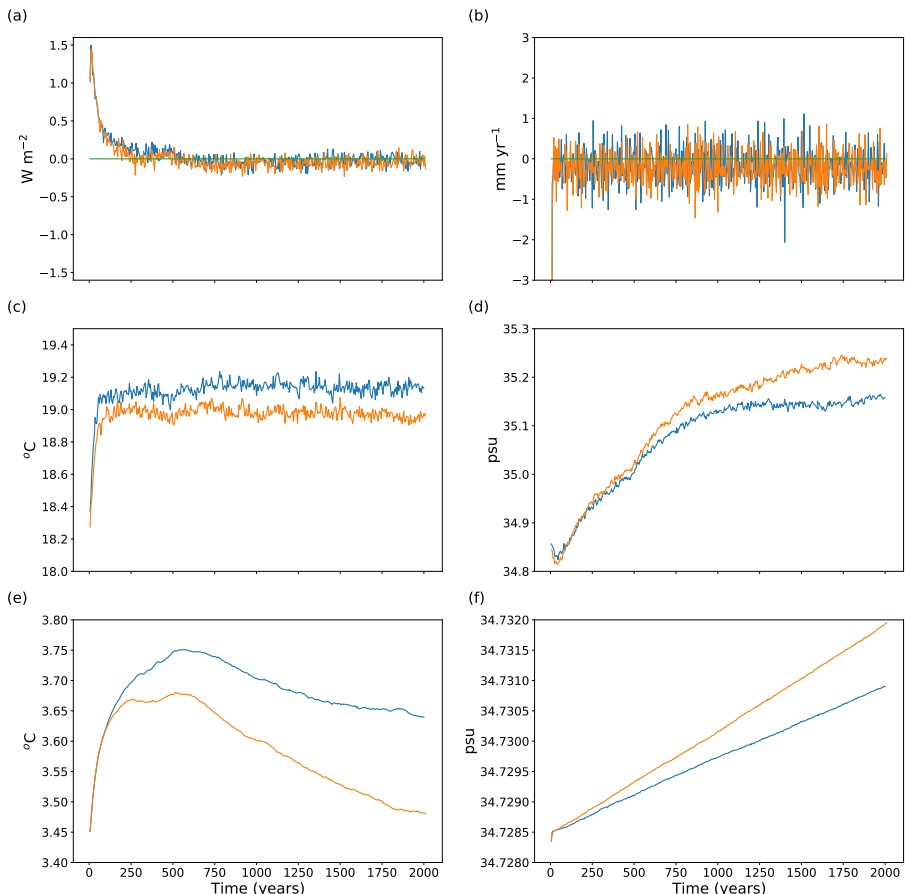

**Figure 1.** Time series of global mean a) surface heat flux (W m$^{-2}$) and b) surface water flux (mm/year) into the ocean, c) SST (°C) and d) SSS, and volume averaged e) potential temperature and f) salinity. Cyan and orange lines show the quantities for the simulations run with the L35 and L20 configurations, respectively.

warms by 0.3°C for L35 and 0.2°C for L20 over the first 500 years, and then cools more steadily, at a rate of approximately 0.005°C/century for L35. After 2000 years the volume average temperature in the L20 configurations is close to the initial value. Salinity shows a gradual trend of 0.000125 PSU/century after an initial adjustment) in L35, which is a reflection of the small imbalance in the fresh water fluxes. The trend in the L20 configuration is around 30% stronger.

5    Global averaged time series of temperature and salinity as functions of latitude and depth are presented in Fig. 2. The time-latitude plots show an initial strong warming of the Southern Ocean that is not density compensated by an increase in salinity at the same latitudes. The onset of this warming occurs quickly and then remains stable for the remainder of the 2000-year integration. A minimal bias develops in the tropical and equatorial regions. In the northern hemisphere higher latitudes there is a strong cooling, that (partly) coincides with a freshening. The differences in time-latitude evolution of the SST and SSS for

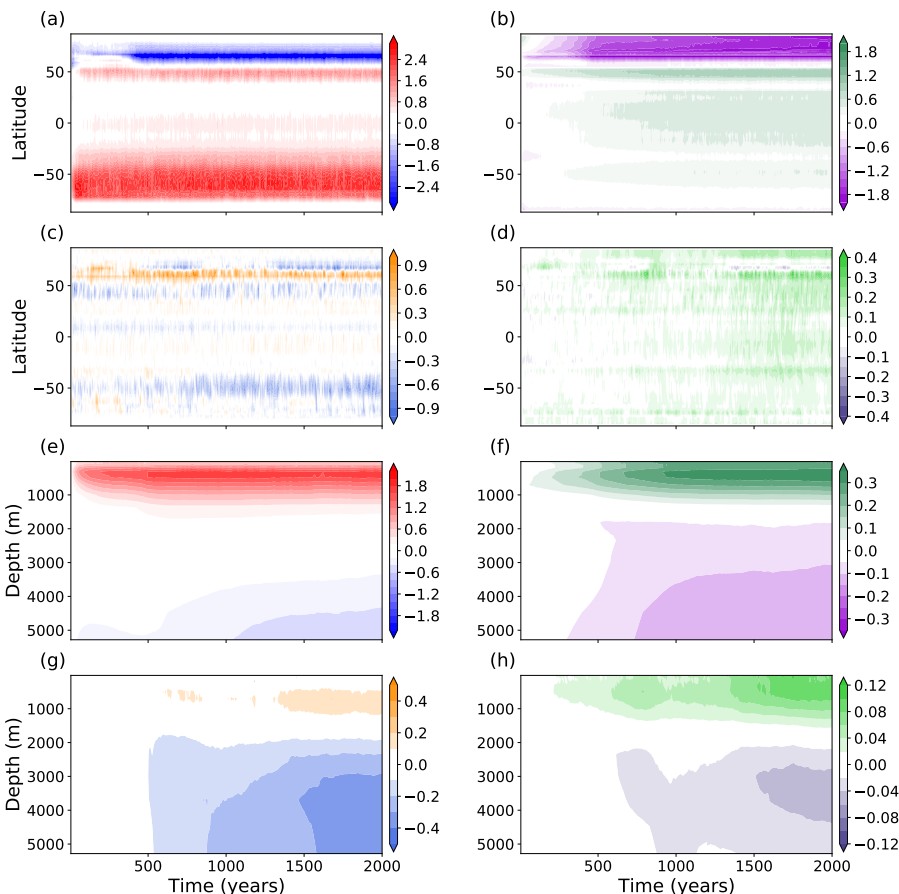

**Figure 2.** Time-latitude plots of drift in annual mean sea surface (a) temperature (°C), and (b) salinity (PSU) in the L35 simulation. Differences (L20-L35) in annual mean sea surface (c) temperature (°C), and (d) salinity (PSU). Time-depth series of global drift in annual mean (e) potential temperature (°C) and (f) salinity (PSU). Differences (L20-L35) in annual mean (g) potential temperature (°C) and (h) salinity (PSU). Drift is relative to initial conditions.

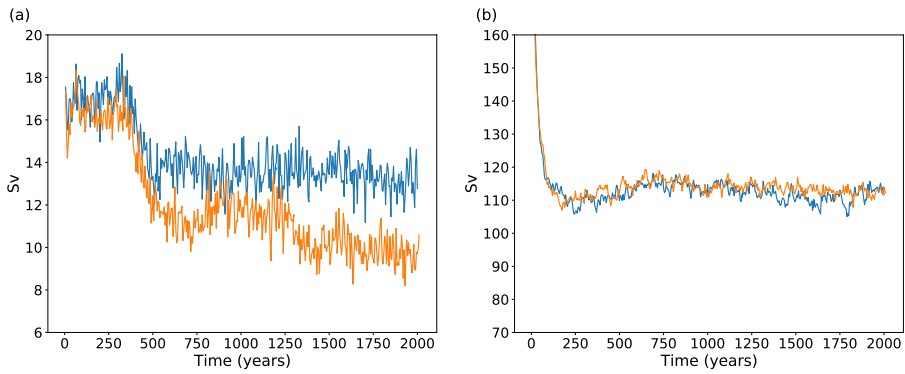

**Figure 3.** Time series of a) the 5-year mean Atlantic MOC, and b) Drake Passage transport in the control integrations. Cyan (orange) lines show transports for the L35 (L20) configuration.

the L20 configuration are shown in Figs. 2 c and d. High latitude SST is slightly cooler in L20 compared with L35, except for a narrow band around 55-65°N where a warming of a few tenths of a degree occurs. Surface salinity is marginally higher and to a large extent latitudinally uniform in L20. The difference in SSS is slightly more pronounced at the same latitude range as the narrow band of warming seen in the SSS. Analysis of the spatial SST and SSS biases presented later in this paper shows

that these anomalies are located in the Nordic Seas. This pattern develops over the first 500 years of the simulation, and then remains stable for the rest of the integration. The SST bias is within the range of that found in the CMIP5 ensemble (Flato et al., 2013).

The time-depth series of potential temperature (Fig. 2 e) compares reasonably with those of other, higher resolution, models such as HadCM3, HadGEM1 and CHIME (Fig. 7 in Johns et al. (2006), Fig. 3 in Megann et al. (2010)). FORTE 2.0 warms

above 1500 m, with the maximum difference from observed values reaching +1.6°C between 400 and 500 m depth. At depths below 4000 m the ocean cools initially, with differences from observations at 5000 m reaching -0.2°C. Differences in salinity from observations, again shown as a global averaged time-depth series, are small (Fig. 2 f). Differences of +0.3 PSU occur between 300 m and 600 m, whilst below 1500 m they are negative and reach a maximum of -0.15 PSU in the abyssal ocean. Differences between L20 and L35 are shown in Figs. 2 g and h. L20 exhibits a stronger cooling trend in the deep ocean, such

that below 2000 m it is around 0.3° cooler than L35 after 2000 years. The dipole structure that develops in the time-depth plot of salinity (Fig. 2 f) is more pronounced, with the biases at 500 m and 3500 m both 25 % stronger in L20 than in L35.

The time series of the maximum AMOC at 30°N is presented in Fig. 3. During the first 300 years of integration a relatively stable 16-18 Sv is maintained in both the L35 and L20 configurations. After 300 years both configurations undergo a reduction, with the AMOC in L35 reducing by ∼4 Sv to 13-15 Sv. The AMOC in L20 reduces initially by ∼4 Sv and then again around

1250 years into the integration, settling at a value around 10-11 Sv. The weaker AMOC in L20 is consistent with the cooling and freshening trends seen at depth, since a weaker AMOC allows a greater influence of Antarctic Bottom Water (AABW) in the abyssal ocean. The AMOC values in both simulations are maintained for the remainder of the 2000 year integrations and

are within the range of those from the CMIP5 ensemble (Heuzé et al., 2015). The standard deviation of the AMOC based on monthly mean values is 3.5 Sv, which is in reasonable agreement with the magnitude of observed variability (McCarthy et al., 2012). Other than the adjustment in the first 500 years, there is little evidence of emergent decadal or multidecadal variability over the course of the L35 control integration, the peak-to-peak range over the last 1500 years of the integration being 3-4 Sv. The initial strong AMOC followed by a reduction after a few centuries is a common feature of FORTE integrations, and appears to be linked to a developing fresh bias over the GIN Seas (shown later).

Transport through Drake Passage weakens rapidly from an initial value of > 160 Sv and from year 200 it settles around 110-115 Sv. This is lower than the recent observation-based estimate of 173 Sv (Donohue et al., 2016) or previous estimates values of around 130-140 Sv (e.g. Cunningham et al., 2003), but within the range seen in other coupled climate models (Beadling et al., 2020). Kuhlbrodt et al. (2012) show that the strength of the ACC correlates with the choice of GM thickness diffusion, with lower values of $\kappa$ yielding stronger ACC transports. Both the L35 and L20 simulations use a value of $\kappa = 2000$. FORTE2.0 has a stronger ACC than other models that use $\kappa = 2000$ (see Fig 1b of Kuhlbrodt et al., 2012). It is closer in strength to other models using values of $\kappa = 700 - 1000$.

## 4    The control climate

After the 2000 year spinup, the frequency of output was increased to monthly and a 25 year long integration was performed. In this section the control climate during this 25 year period is presented.

### 4.1    The Atmosphere

Annual time mean surface air temperatures (SATs) in the tropics are 25°C, with some regions over land reaching 30-35°C. The Arctic reaches -20°C, with temperatures over Greenland reaching as low as -40°C and the interior of Antarctica reaches as low as -60°C (Fig. 4 a). Anomalies of annual mean SAT from the early years (1871-1896) of the twentieth century reanalysis (20CR Compo et al., 2011) are presented in Fig. 4 b. FORTE 2.0 performs well at low latitudes, especially over the ocean. There is a pronounced warm anomaly of around 5 °C over most of the Southern Ocean, consistent with the positive SST bias shown in section 4.2. A cold SAT anomaly with respect to 20CR also manifests over the Nordic Seas, extending into the Barents Sea. The L20 simulation exhibits very similar annual mean SATs (not shown). Values for L20 are within 1 °C of those shown for the L35, except for two regions over the northern Labrador Sea and the Weddell Sea which are cooler in L20 by several degrees. These differences are likely due to the difference in the response of the flux barrier which represents sea ice. The SAT seasonal range for FORTE 2.0 and 20CR (1871-1896) are presented in Fig. 4 c and d. Seasonal SAT variability over the tropical ocean is low, whilst variations in SAT over land are much higher, reaching 40°C over the Eurasian continent. The seasonal range of FORTE 2.0 compares reasonably well with 20CR throughout most of the low and mid-latitudes. However, the seasonal ranges in SAT at high latitudes (both Arctic and Antarctic) are much larger than those in 20CR. Arctic and Antarctic seasonal SAT variability is 40-45°C, with the coldest regions of Antarctica reaching as low as -85°C during July.

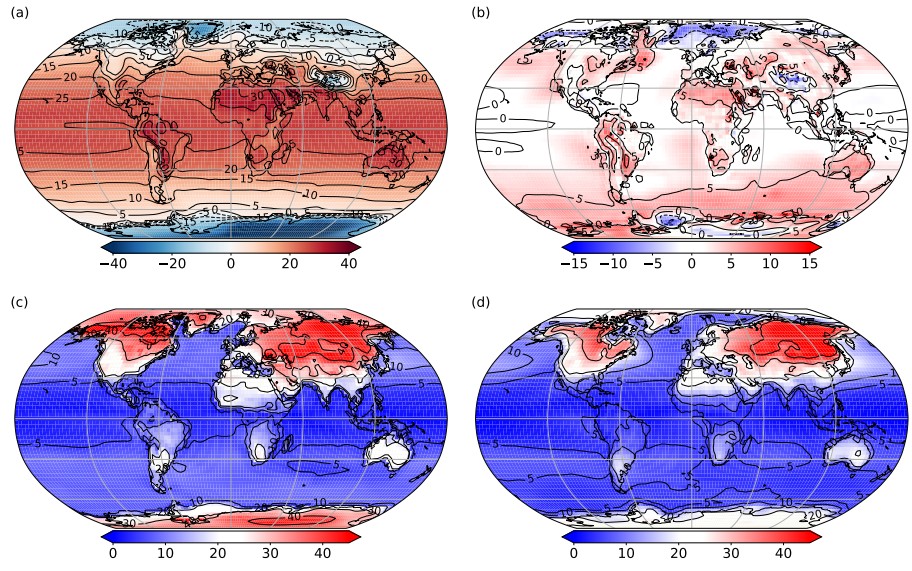

**Figure 4.** Time mean (years 2001-2025) plots from the L35 configuration of: surface air temperature (°C) for a) annual mean, b) anomaly from 20CR years 1871-1896, c) seasonal range in FORTE 2.0 and d) seasonal range in 20CR years 1871-1896.

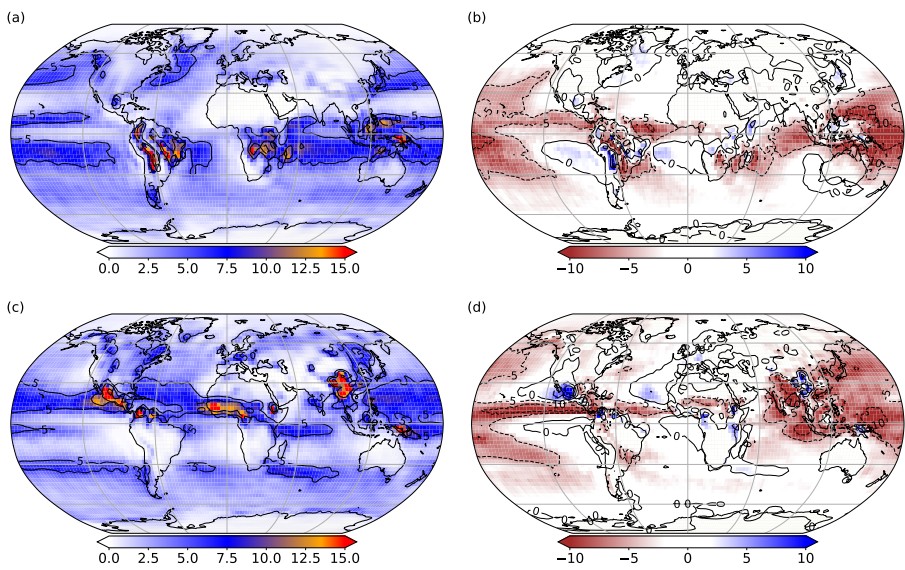

**Figure 5.** Time mean (years 2001-2025) plots from the L35 configuration of: precipitation (mm day$^{-1}$) for a) DJF, b) DJF anomaly from 20CR years 1871-1896, c) JJA, and d) JJA anomaly from 20CR years 1871-1896.

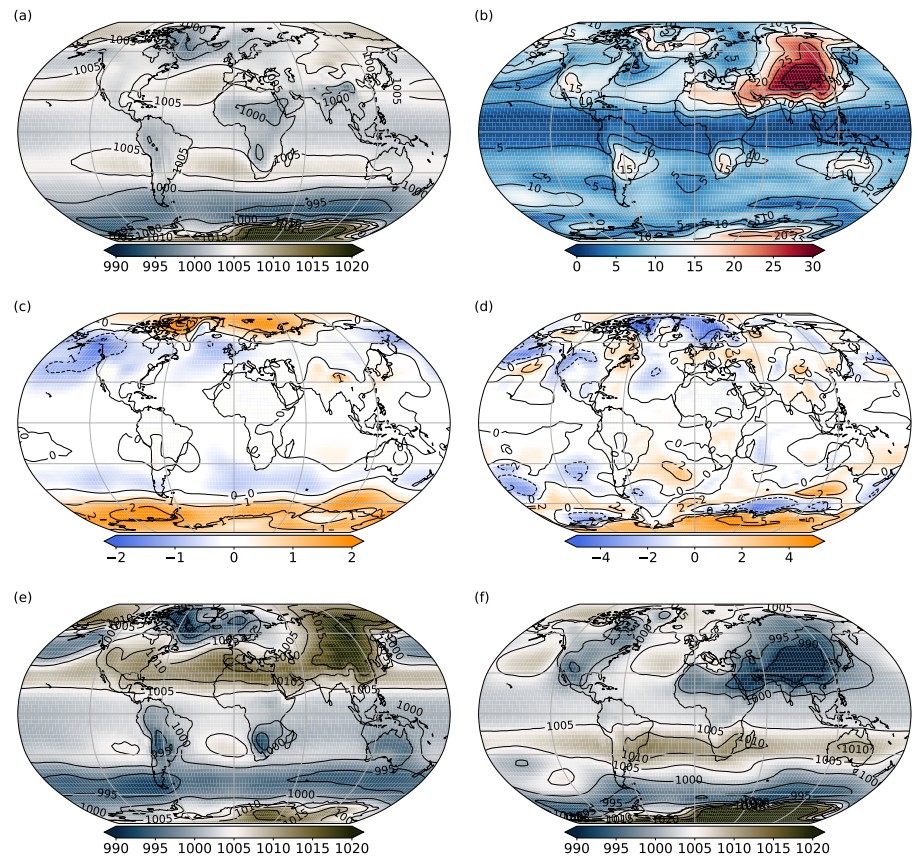

**Figure 6.** Time mean (years 2001-2025) plots of: sea level pressure (mb) for a) annual mean, b) seasonal range, c) difference in slp L20-L35, d) difference in seasonal range (L20-L35), plus values for the months e) January, f) July from the L35 simulation.

Fig. 5 shows winter (DJF) and summer (JJA) mean precipitation and anomalies with respect to the 20CR 1871-1896 climatological mean. During DJF (Fig. 5 a), regions of high precipitation over the northern hemisphere oceanic western boundary currents (up to 6 mm/day and extending to northwestern Canada and along the Gulf Stream and western boundary current track in the north west Atlantic) are evident, as well as high values (10-12 mm/day) over tropical Africa and South America, Indonesia and over the Inter Tropical Convergence Zones in the Atlantic and Pacific Oceans. Very low values (0-1 mm/day) are seen over the polar regions, the subtropical desert regions (terrestrial and oceanic) and (unrealistically) over the equatorial Pacific. During summer (JJA) the ITCZ and the corresponding high levels of rainfall shift northward (Fig. 5 c). There is enhanced rainfall due to the Asian summer monsoon, though in FORTE 2.0 it does not extend sufficiently far east. The anomalies with respect to the 20CR 1871-1896 climatological DJF mean (Fig. 5 b) show that, particularly over the tropical Pacific, FORTE 2.0 simulates too little rainfall. This is most pronounced over the western tropical Pacific during DJF and also over the northern extent of the ITCZ in the eastern Pacific during JJA (Fig. 5 d). A major difference with the observed

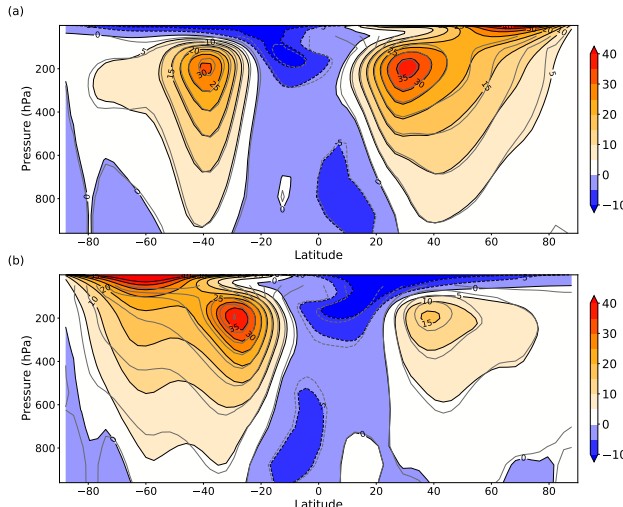

**Figure 7.** Time and zonal mean zonal wind velocities (m s$^{-1}$), for a) Winter (DJF) mean, and b) Summer (JJA) mean. The shading and black contours show the velocities for the L35 simulation, whilst the grey contours show the equivalent contours for the L20 simulation.

distribution is the South Pacific ITCZ, which is narrow and predominantly zonal in the model solution, whereas observations show a broader north-west to south-east oriented region.

Contours of annual average mean sea level pressure, displayed in Fig. 6 a, show the expected bands of high pressure over the subtropical oceans (e.g. the Azores and North Pacific Highs) and over the polar regions and low pressure cells at mid
latitudes (e.g. Icelandic and Aleutian Lows) and over the Equatorial regions. The seasonal range is largest over land (Fig. 6 b), particularly highlighting the seasonal variability over Siberia. Differences in the annual mean slp anomalies and seasonal range (L20-L35) are predominantly poleward of 60°N and S, (Figs. 6 c and d). The seasonal range is smaller in L20 over the Labrador and GIN Seas and over the high latitude Southern Ocean. Contours of sea level pressure show the intensification of the surface winds over the midlatitudes in both southern and northern hemispheres during the winter season (Figs. 6 e and f).
We note that the Siberian high is not very intense for mean January conditions and this appears to have the effect of allowing the Icelandic Low to expand and displace eastwards over Scandinavia, resulting in a displacement of the winter NAO pattern compared with observations (see section 5).

Time mean zonal wind for both summer and winter is shown in Fig. 7 as a function of latitude and pressure. The model exhibits northern and southern hemisphere jet streams at around 40°S and 40°N at 200 hPa. The southern jet stream exhibits a
lower seasonal range (28–36 m s$^{-1}$) than the northern jet stream (12–36 m s$^{-1}$). Surface westerlies and easterlies are of order $\pm$0–4 m s$^{-1}$ in the annual mean. The most notable differences between the L20 and L35 configurations occur during JJA, with weaker winds at 60°S. The mid-latitude cores are also slightly stronger during JJA in L20. At 80 °S the zero contour extends down to the surface, indicating a change in the mean wind direction from weak westerlies to weak easterlies.

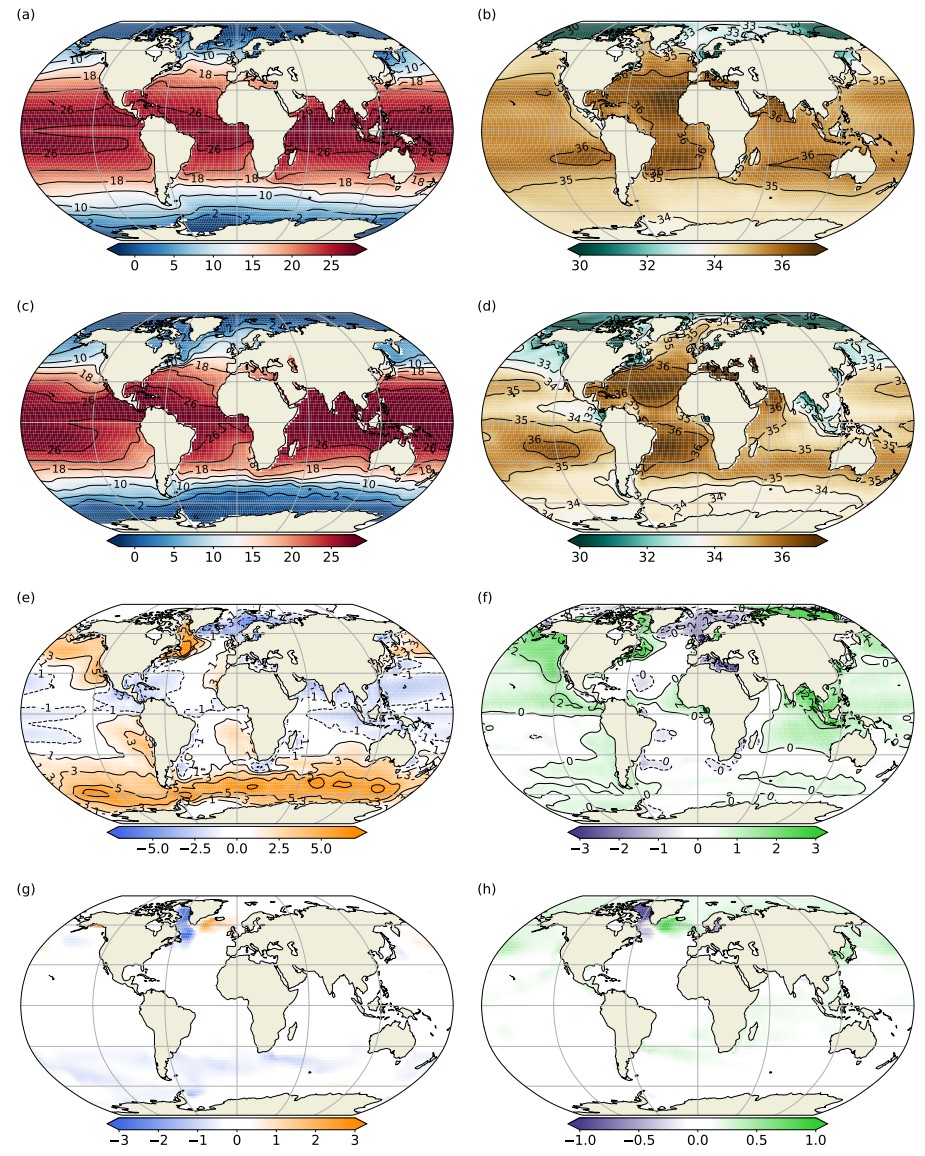

**Figure 8.** Time mean (years 2001-2025) SST and SSS for FORTE 2.0 L35 (a and b), and for EN3 (c and d). Panels e and f show FORTE 2.0 L35-EN3 SST and SSS anomalies, respectively, whilst panels g and h show SST and SSS anomalies for L20-L35.

## 4.2 The Ocean

Annual mean SST (Fig. 8 a) shows maximum temperatures in the Indian and tropical Pacific and Atlantic Oceans reach 26°C. Compared with the EN3 climatology (Ingleby and Huddleston (2007), Fig. 8 c) there is a cool anomaly of around 1°C throughout the tropics (Fig. 8 e). Regions immediately west of the major land masses (coincident with regions of coastal upwelling)

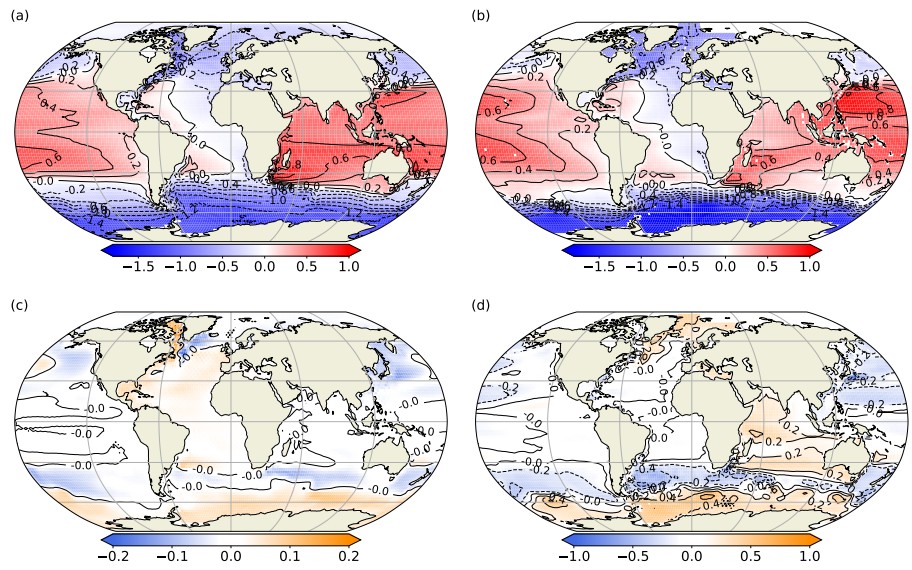

**Figure 9.** Time mean (years 2001-2025) of SSH (m) for a) FORTE 2.0 L35 and b) OCCA (2004-2006) climatology. Differences in SSH for c) L20-L35 and d) L35-OCCA.

show warm SST errors of 2-3°C magnitude, probably arising from a known issue in many coupled climate models related to the poor representation of marine stratocumulus cloud (Gordon et al., 2000). There is a substantial warm bias throughout the Southern Ocean and extending into the southern parts of the Pacific and Indian oceans, likely due to a combination of deficiencies in the physical representations of the ocean dynamics and cloud physics (Hyder et al., 2018). The Nordic Seas are

5 several degrees cooler and up to 1.5 psu fresher (Fig. 8 f) than EN3, possibly due in part to the crude representation of sea ice, and in part due to the inadequate representation of ocean circulation in the Arctic and Nordic Seas in a 2° resolution ocean model. There is a positive salinity bias of around 3 psu further east in the Arctic, north of Siberia. Although large, the size of the salinity bias in the Arctic is not uncommon, even for models that do not require a polar island to prevent issues arising from the convergence of the grid at the north pole (Megann et al., 2010). Annual mean SSS is well represented throughout the

10 southern hemisphere ocean, where errors are mainly confined to within +/- 0.5 psu (Fig. 8 f). Positive biases of order 1-1.5 psu occur in the Bay of Bengal and around the maritime continent and the northeast Pacific. The Labrador Sea and the region extending along the US coastline as far south as Cape Hatteras shows shows positive salinity biases between 0.5 and 2 psu, the latter coincident with a positive temperature bias that exceeds 5°C in a small region is indicative of the Gulf Stream separating too far north, bringing tropical waters too far north and west. Worth noting, though, is that the L20 simulation exhibits smaller

biases in both SST and SSS in the Labrador and Irminger Seas (Fig. 8 g and h). There is also a slight improvement in the Southern Ocean warm bias in L20 compared with L35. Some of the model biases will arise from the relatively coarse horizontal and vertical resolution and missing physical processes. However, as indicated by the differences between the L20 and L35

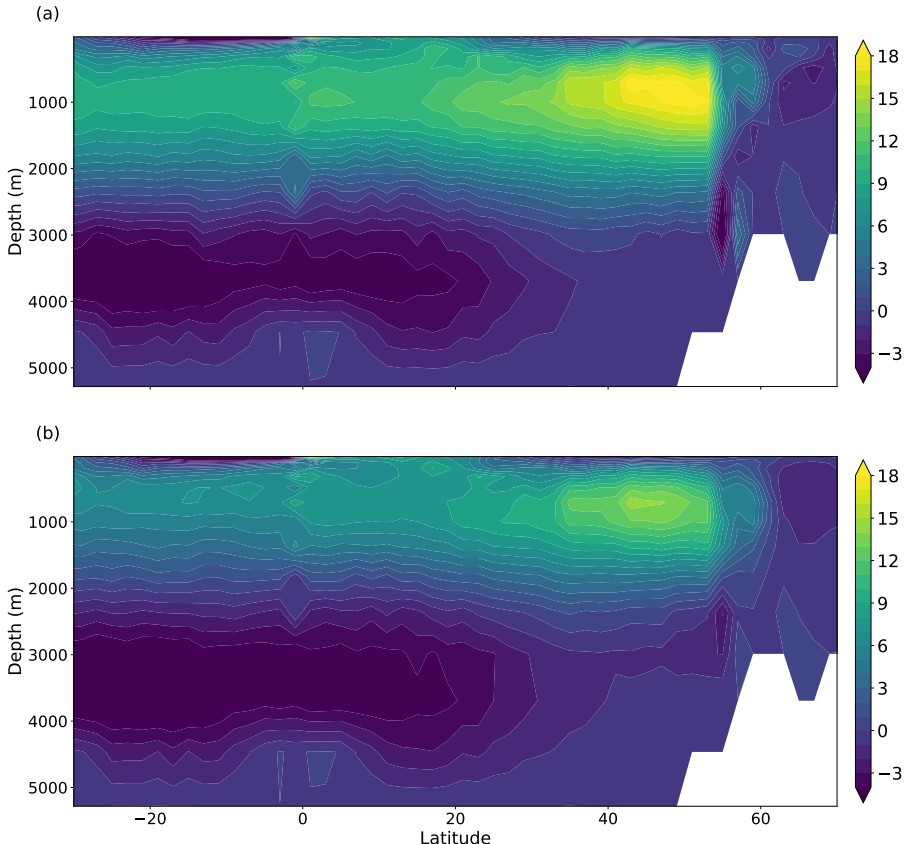

**Figure 10.** AMOC (Sv) as a function of latitude and depth, averaged over the years 1900-2025 of a) the L35 integration, and b) the L20 integration.

simulations, it is likely that a substantial reduction of biases would be achieved with the application of a rigorous calibration methodology such as History Matching (Williamson et al., 2014).

Sea surface height provides insight into the wind driven ocean circulation. The SSH from L35 and the OCCA climatology are shown in Figs. 9 a and b, respectively. Gyre circulation in all the major ocean basins is highlighted by the contours, along

5    with regions of intensified flow, such as the Gulf Stream, the Kuroshio, and along the northern boundary of the ACC. However, the coarse resolution of the ocean model results in flows that are too broad and diffuse, weakening the SSH gradient across these intensified flows. The North Atlantic subpolar gyre appears constrained to the west of the basin. Slumping of the SSH gradient across the ACC is evident in the anomaly of L35 with respect to OCCA (Fig. 9 d), and corresponds to the weak ACC transport shown earlier (Fig. 3). The slope in SSH is also weaker in the N. Atlantic and extending into the Nordic Seas.

10   Comparison with L20 (Fig. 9 c) shows a slight steepening of the gradient across the ACC in L20, a small reduction in the bias compared with the OCCA climatology.

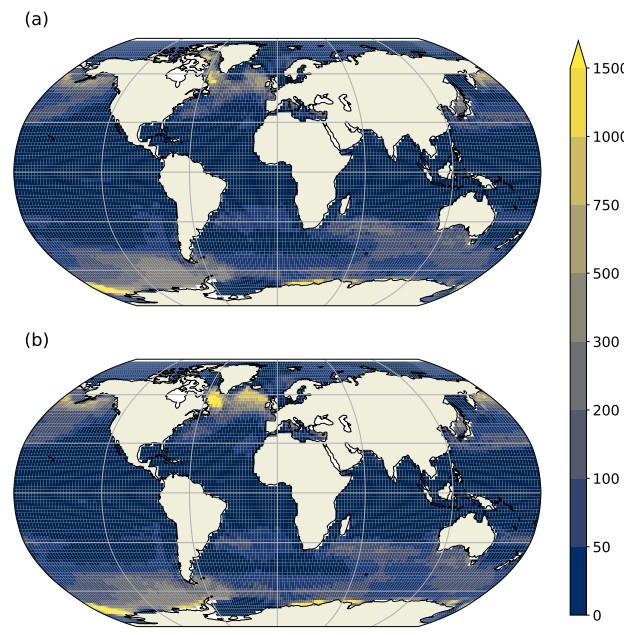

**Figure 11.** Winter mixed layer depths (m) averaged over the years 2000 - 2025 of a) the L35 integration, and b) the L20 integration. The northern(southern) hemisphere shows mixed layer depths for the month of March(September). The mixed layer depth is defined as the depth at which a density difference from the top layer of 0.03 kg m$^{-3}$ occurs.

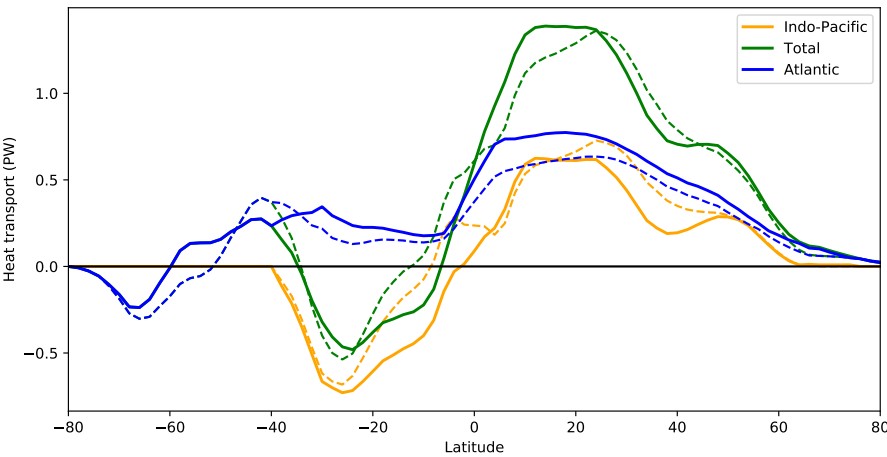

**Figure 12.** Meridional heat transport (PW) as a function of latitude for the Global ocean, Atlantic and Indo-Pacific, averaged over the control integration. A 5-grid point smoother has been applied. Solid (dashed) lines show the meridional heat transport for the L35 (L20) simulations.

A latitude depth plot of the AMOC shows a maximum around 50°N and at 1000 m depth (Fig. 10). Closely packed streamlines at the high northern latitudes indicate that much of the deep convection occurs abruptly in a narrow latitude band and southward North Atlantic Deep Water transport reacehss around 2.5 km depth. The abrupt sinking at the high northern latitudes is characteristic of coarse resolution ocean models where flow into the Nordic Seas is poorly represented. Winter mixed layer depths in the southern Labrador Sea reach 2500 m in a few grid cells, whilst winter mixed layer depths south of the Denmark Strait, Iceland and the Faroe Bank Channel can reach 1000 m (Fig. 11). Wintertime convection is too shallow in the Nordic Seas, with mixed layer depths reaching 125-150 m in the central and eastern Nordic Seas. The AMOC transport through 30°S is 10 Sv in L35 and 6 Sv in L20, and is stronger (~14 Sv (L35), ~10 Sv (L20)) at 30°N. There is a strong AABW cell (~6 Sv) centered at 3500 m depth, which weakens to about 2 Sv at 30°N. As mentioned earlier, the AABW cell in L20 is slightly stronger, and in Fig. 10 is shown to extend further north. There is evidence of two grid point noise at the Equator, which has been identified previously in Bryan-Cox models (Weaver and Sarachik, 1990). The structure of the AMOC is similar in both the L35 and L20 simulations, with the L20 configuration consistently around 30% weaker.

Ocean meridional heat transport (OHT) in FORTE 2.0 is around 60% that expected based on observational estimates, but consistent with the weaker than observed volume transport (Fig. 12). Atlantic OHT at 26°N is 0.74 PW in L35 and 0.63 PW in L20, whilst observationally derived estimates suggest the current value is closer to 1.3 PW (Johns et al., 2011). Globally, the OHT reaches 1.4 PW, instead of the $2.1 \pm 0.3$ PW computed by Trenberth and Caron (2001). Over the Southern Ocean (35-65°S) OHT is northward, a characteristic seen previously in MOM-based ocean models (de Freitas Assad et al., 2009). This may be related to the strong warm SST bias present over region 40-60°S (Fig. 8) and its consequent effect on surface heat fluxes.

## 5  Modes of variability

A primary aim for any climate model is to adequately reproduce observed modes of climate variability sufficiently well that the model can be used to study the observed phenomena in a variety of contexts. In this section we present analysis of some of the most important modes using monthly mean ocean output for the years 1600-1950 of the control simulation and daily surface pressure output during years 1600-1699 of the control integration.

Composites of the SST anomaly during El Niño and La Niña years show the spatial pattern of the anomalies throughout the tropics (Fig. 13 a, b). Both phases of ENSO are weaker than observed, in particular nearer the eastern boundary. The composite temperature anomaly reaches a maximum of 1°C for the region 160°W-100°W, 5°S-5°N, whilst the characteristic region of observed strong SST anomalies near to the coast of central and southern America only reaches 0.7°C and is not strongly connected to the warm anomaly in the central Pacific. This is probably related to the fact that the South Pacific Convergence Zone is too zonal and extends all the way across the Pacific, which is a common feature in coupled climate models (Niznik et al., 2015). The time series of temperature anomalies in the Niño 3.4 region shows a number of strong temperature anomaly events, although the magnitude is in general too small (Fig. 13 c, d). We plot the distribution of SST anomalies for the Niño 3.4 region for both model configurations and for HadISST data for the period 1870-2019 (Rayner et al., 2003; Trenberth,

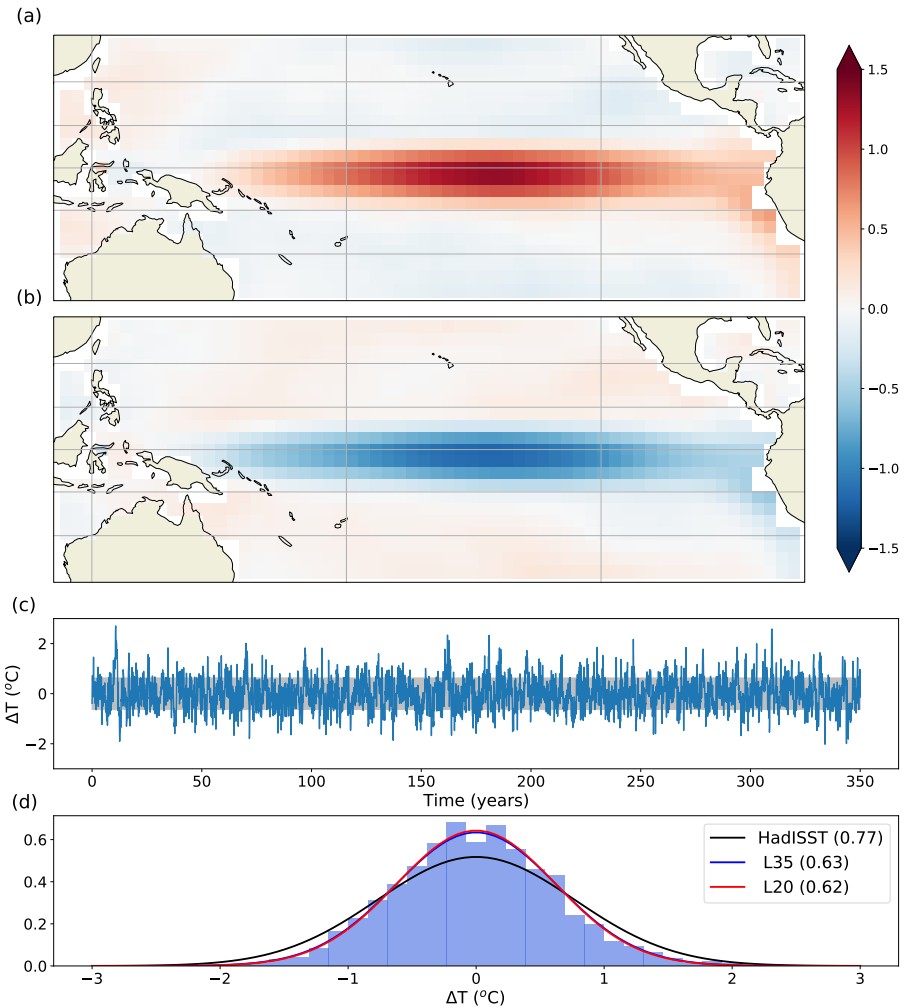

**Figure 13.** Composite anomaly of a) El Niño events, and b) La Niña events from the L35 simulation. ENSO events are defined as those which exceed ± 1 standard deviation anomaly within the Niño 3.4 region. c) SST anomaly time series and d) Histogram of SST anomaly distribution relative to the mean for years 1601-1950 from the L35 simulation. The Gaussian curves in d are fits to the distribution of Niño 3.4 SST anomalies for HadISST (black, Trenberth (2020)), L35 (blue), and L20 (red). The blue and red lines are very close and the red line mostly overlies the blue line. Standard deviations are given in brackets.

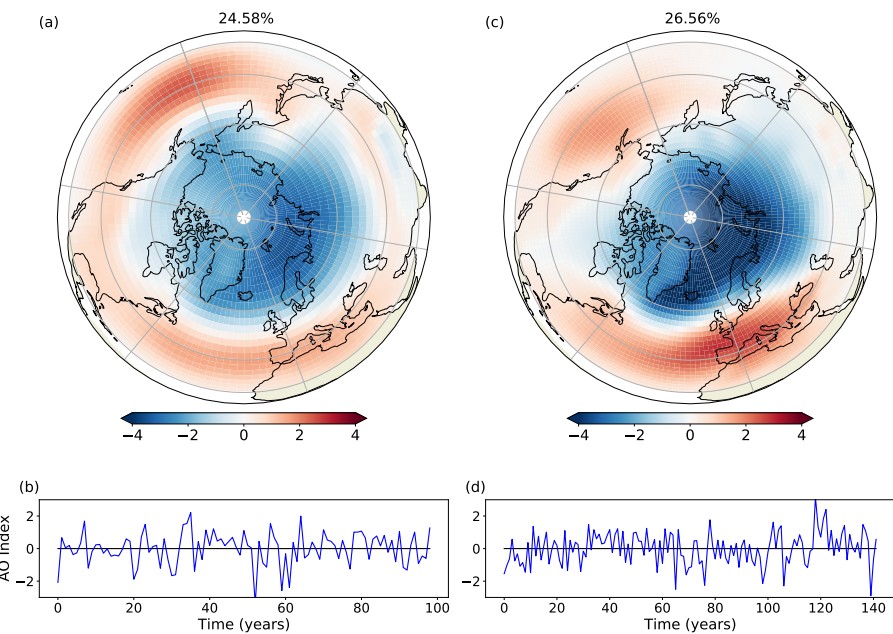

**Figure 14.** The AO as represented by the first EOF and PC computed using deseasoned and latitude weighted sea level pressure for FORTE 2.0 (a,b), and 20th Century Reanalysis (c, d)

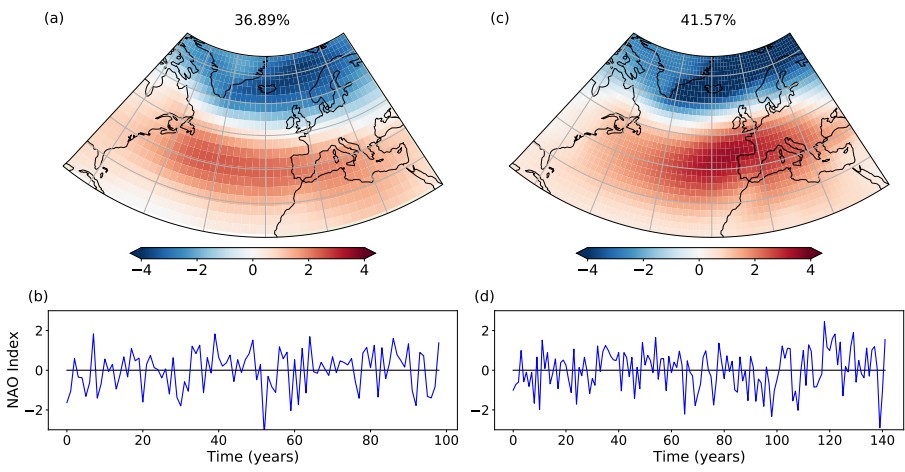

**Figure 15.** The NAO as represented by the first EOF and PC computed using deseasoned and latitude weighted sea level pressure for FORTE 2.0 (a, b), and 20th Century Reanalysis (c, d)

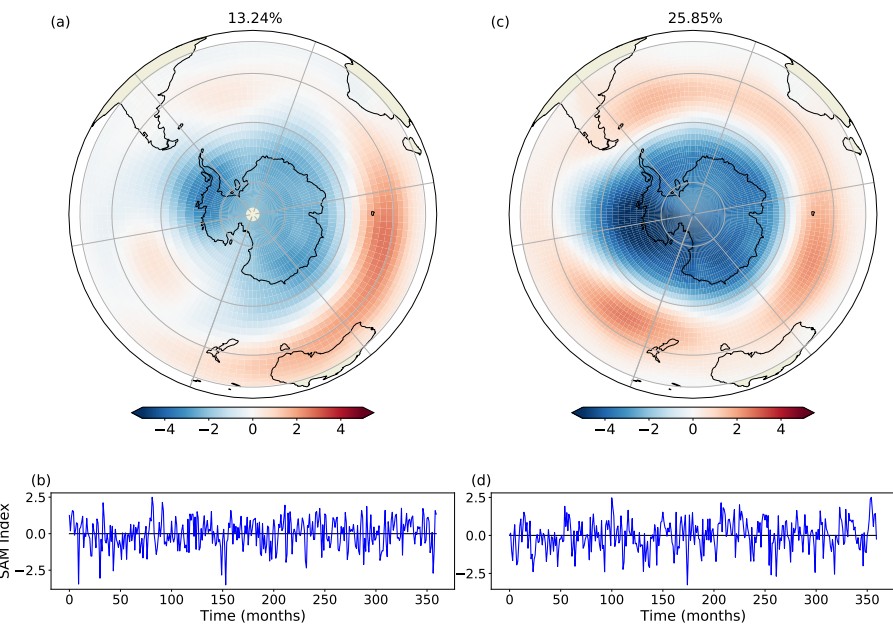

**Figure 16.** The SAM as represented by the first EOF and PC computed using deseasoned and latitude weighted sea level pressure for FORTE 2.0 (a, b), and 20th Century Reanalysis (c, d)

2020). The distribution of the histogram is too narrow compared with observations (Fig 13 d), and there is very little difference between the distributions for the L20 and L35 simulations. In both model configurations the extreme values extend to around ±2°C (Fig 13 d)), whilst observations suggest the extremes should be closer to ±2.5°C.

We also examine the main extratropical modes of variability predicted by FORTE 2.0 in the northern hemisphere. We
compare 20°-90°N area weighted empirical orthogonal function (EOF) analysis of the deseasoned and latitude weighted sea level pressure fields from FORTE 2.0 and, for comparison, the 20th Century Reanalysis (Compo et al., 2011). FORTE 2.0 produces an annular mode structure as the main mode of variability, corresponding to the Arctic Oscillation in observed data. In agreement with observations (e.g. Thompson and Wallace, 2000; Ambaum et al., 2001), the model reproduces the two mid latitude centres of action over the North Pacific and North Atlantic, with the Pacific centre stronger and the Atlantic centre
slightly weaker than those seen in the 20th Century Reanalysis and the locations of their maxima displaced westward towards the western half of each ocean basin (Fig. 14). The strength of the Arctic pole in FORTE is also weaker than observations. The North Atlantic Oscillation (NAO) is closely related to the AO and is one of the principal modes of atmospheric variability in the North Atlantic sector (Hurrell, 1995). We compute area weighted EOFs for the NAO over the region 20°-80°N, 90°W-40°E. The first EOF and its accompanying principal component are presented in Fig. 15. In the North Atlantic there is a good
approximation to the NAO pattern, but in FORTE 2.0 the centre of the southern lobe is displaced westward and the northern

lobe extends further south over mainland Europe compared with the observed pattern. Again, the principal component suggests more high-frequency variability in observations than FORTE 2.0.

Similar to the northern hemisphere, the Southern Annular Mode, or Antarctic Oscillation represents the principal mode of climate variability in the southern hemisphere. Here we compute area weighted EOFs over the region 20°-90°S. FORTE performs less well in the southern hemisphere (Fig. 16), with the annular structure significantly weaker over the Pacific and Atlantic sectors. The variance explained by the first EOF is also greatly reduced in FORTE, approximately half that seen in the 20th Century Reanalysis, and is likely to be linked with the anomalously warm Southern Ocean SST.

## 6  Summary

We present an assessment of two 2000 year simulations of the FORTE 2.0 coupled climate model, one using the 35 $\sigma$ layer atmosphere including a stratosphere (L35), and one using the 20 $\sigma$ layer atmosphere without a stratosphere (L20). The model integrates from rest and is sufficiently fast to enable studies of multi-centennial climate variability. The model is economic to run, and can be adapted and configured to study a wide range of climate questions.

The simulations presented here are not optimally tuned for any specific purpose, but our assessment indicates that FORTE 2.0 is able to simulate a satisfactory climate state and climate variability. Biases that develop in the mean state are comparable to those found in other coupled climate models (Flato et al., 2013) and particularly those of similar complexity and resolution. A small imbalance in the fresh water budget (see Fig. 2) would need to be addressed for studies extending over timescales much longer than several millennia. Modes of climate variability in the northern hemisphere are represented well, though there are shortcomings in the southern hemisphere variability that are likely related to a strong SST bias over the Southern Ocean. Identifying the cause(s) of such biases is often a complex process in itself (Hyder et al., 2018), and beyond the scope of this current work. A further step would be to rigorously calibrate the model to improve the simulated climate and to better understand the limitations and behaviour of the modelled climate system.

## 7  Code availability

The code, compilation instructions and example run scripts, together with all necessary ancillary files are accessible at: http://doi.org/10.5281/zenodo.3632569. The configuration committed to the Zenodo archive is the one used to produce the simulation presented in this paper. Processing of the IGCM4 output requires the program BGFLUX, a copy of which is accessible from the FORTE2.0 GitHub repository linked to the Zenodo archive. A comprehensive user guide/manual for FORTE 2.0 does not currently exist. A folder titled Documentation has been added to the FORTE2.0 GitHub repository, and this contains relevant references and copies of technical documents from the original FORTE and component models.

## 8  Data availability

The code and data required to reproduce the figures presented in this paper are provided in a supplementary file.

*Author contributions.* ATB, MJ and BS developed the coupled model configuration from versions used in earlier studies. The original coupling of FORTE was performed by BS and RSS. All authors were involved in finalising the configuration presented here. MJ undertook all model simulations. ATB wrote the paper, analysed the output and prepared all tables and figures. All authors edited the paper text.

*Competing interests.* The authors declare that they have no competing interests.

5  *Acknowledgements.* We acknowledge the support of resources provided by the High Performance Computing Cluster supported by the Research and Specialist Computing Support service at the University of East Anglia. We also acknowledge the support and contributions from colleagues on earlier versions of the code and simulations, most notably M. Brand, C. Wallace, and M. Stringer. We gratefully acknowledge the developers of the software packages PVM and OASIS2.3, both of which are instrumental in the coupling for FORTE 2.0. We thank the editor and reviewers for their contributions. We also acknowledge William Dow, who retrieved and installed FORTE 2.0 from the repository

10  and provided useful feedback to improve the installation instructions. ATB, JJMH and BS received support from ACSIS (NE/N018044/1); BS was also supported by SMURPHS (NE/N005686/2). MJ was partially funded by SMURPHS (NE/N006348/1).

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
