# Peer review of "FORTE 2.0: a fast, parallel and flexible coupled climate model"

_Geoscientific Model Development, 2020_

## Referee Comment (RC1) · Anonymous Referee #1 · 1 Apr 2020

The authors present an intermediate resolution coupled Ocean Atmosphere General Circulation Model that has been used previously but for which no proper model description is available. It is thus certainly worth to briefly describe the model and its mean state here. However, the authors fail in highlighting the strengths of the tool compared to similar intermediate resolution models and the description of the model and of its results are too superficial to be really useful. I thus consider that major modifications are required before the final publication of the manuscript.

1/ As it is presented, FORTE2.0 is an intermediate resolution model like many others. The fact that the model is relatively fast does not appear clearly as an advantage compared to similar models that should be more or less as fast if they are run at the same resolution. If I understand well, the tool seems to have two main advantages. First, the

model can include or not the stratosphere. However, the stratospheric dynamics is not at all discussed. The minimum for me would be to present simulations with and without resolved stratosphere and see the impact of this choice on model result. 2/ It seems that the model is readily configurable, providing maybe more flexibility than other tools. This interesting aspect is mentioned but not developed enough to see if this is a real strength of the tool. The manuscript should thus insist much more on those potential strengths (and maybe on others).

2/ The model evaluation is very general. Many times it is mentioned that the results of the model are satisfactory or within the range of other models but the model performance itself is not really quantified. The observations are shown for some quantities for comparison with model results (some figures in section 4.2, figures 12-14) but not for many others (section 4.1, figure 14). This makes the evaluation harder to follow. Furthermore, it is not clear from the text if the simulation presented is from a 'standard configuration' of the model that may be used as a future reference or just an illustrative version that is not supposed to last and will not be used later. If the configuration presented is not a standard one, this strongly diminishes the interest of any diagnostic performed with this configuration and I would recommend that such a standard version is obtained before presenting it.

3/ The model has no interactive sea ice. This is a strong limitation compared to similar tools and this should be mentioned earlier (in the abstract for instance) as this may be an important element for potential users.

Specific comments

Page 1, lines 16- 21. The distinction between 'coarse resolution simplified models', 'intermediate resolution models' and the 'Earth Models of Intermediate Complexity (EMICs)' (introduced page 2, line 30) is to very clear to me. For instance, I would personally put ECBilt in the list of EMICs, and thus among the 'coarse resolution simplified models' rather than within the 'intermediate resolution models'.

[Figure]
Even if it is always better to use the most up-to date datasets, I do not think that using relatively old ones like the climatology of Levitus and Boyer (1998) - Levitus et al. (1998) (page 2, line 15) or the heat transport of Trenberth and Caron (2001) (page 13, line 18) makes a big difference but stopping the analyses in 1997 for ENSO (page 16, line 11) may seem a bit too early to have a good evaluation.

Page 3, line 28. It is not clear if the variable drag coefficient is applied both over land and ocean.

Page 3, line 34. Not clear to me what is meant here by 'ice has melted' if there is no representation of sea ice

Page 4, line 9. What is the size of the 'polar island' ?

Page 11, Figure 3. What is EN3? Not sure it is defined.

Page 13, Lines 9-14. Please specify where convection occurs in the model. The wording 'not uncharacteristic of coarse resolution ocean models' is another example of a general sentence where more substantial, quantified information would be required.

Page 16, line 11. The evaluation of ENSO characteristics is based on a figure from a paper published in 1997 while it would be very easy to evaluate precisely the simulated Nino3.4 index compared to observed one.

Page 17, line 6. It is mentioned that '. Comparison of the corresponding principal component time-series (Fig. 12) suggests the presence of some higher frequency variability in observations that is not captured by FORTE 2.0'. This should be quantified both for the AO and the NAO.
* * *

---

## Referee Comment (RC2) · Dmitry Sidorenko (Referee) · 27 May 2020

The paper describes and evaluates a fast, parallel and flexible coupled climate model (FORTE2.0) which due to its simplicity but still retaining the realistic physics of the fluid flow, can be run with a noticeable throughput of ∼100 model years per day on on 28 cores. Possibility of running FORTE2.0 on a desktop computer makes it an ideal model for the academic studies and climate research. The paper demonstrates that FORTE2.0 produces a stable climate and in terms of certain performances is not worse than most of high resolution CMIP models. The description paper for FORTE 2.0 is certainly needed and the GMD is the proper journal for this.

The paper is good written optimally organized, explains what FORTE 2.0 is about and

is ready for publishing after minor corrections. I have several comments which the authors may wish to elaborate on in the revised version of the paper.

A general question before downloading the code: in which programming language is FORTE written?

Page 3, lines 2 and 3: It sounds like two different grids are being used in the atmosphere. I would therefore rephrase to something like "A longitudinally regular and Gaussian in latitude grid with a grid spacing of ~2.8° is used for advection and diabatic processes."

Page 3, line 31 and below: "FORTE 2.0 does not include dynamic sea-ice representation. Instead, sea ice is represented by a barrier..." Do I understand this correctly that there is no dynamic sea ice nor the sea ice itself but the flux barrier? I am curious about the process of flux computation in this place. How the values for temperatures and albedo to parameterize the presence of sea ice were chosen? At the end of the page the authors say "...until the ice has melted". Considering what is said above, is it the same as ... until the atmospheric temperature becomes > 271°K?

I assume the restoring of SST below the sea ice a part of the ocean component? Maybe it is worth mentioning this since this chapter describes the atmospheric component.

Page 4, line 13: How the topography was interpolated? Was it smoothed in between or not? conservatively?

Page 4, lines 18 to 20: Considering the model "biases" which are shown below in the paper I wonder whether the geometrical scaling of GM could improve the solution? Was there a run made without GM? Does it improve the SPG in the NA?

Page 5, lines 2 to 3: in the Table 1 the background vertical diffusivity is defined as a constant value but it is stability dependent here. Which mixing scheme is used above the background?

Page 5, section 3: it is worth repeating that the pre-industrial atmospheric concentrations of CO2 were used.

Page 5, line 16: should there be the minus sign? The ocean warms initially but cools towards the end of the control run. Actually most of models simulate higher than observed ocean temperatures even under pre-industrial forcing (e.g. Griffies et al. 2011, doi:10.1175/2011JCLI3964.1; Lucarini and Ragone 2011).

Page 5, line 17: is the salinity trend caused by the use of the linear free surface (W_surf * SSS)?

Page 6 line 8: the drop in AMOC happens abruptly. Do you have any idea of what has happened?

Page 8 line 5: Is there a link between low ACC and GM (or is it because of winds)?

Page 8 line 9: I wonder why only 25 years? Do you expect any change in results if a longer period is considered?

Page 10: A pre-industrial run is compared with the present day climatology. Bias seems to be not the proper word then. Maybe it is also worth mentioning that the Levitus climatology is being used as a metric only.

Page 11, line 13: the commonly observed "cold bias" around Newfoundland is replaced by a warm anomaly instead. Something very different from most of the climate models is happening there. Same in SSH (page 13, line 5). Is it because of the winds? Does FORTE depict any MLD in the Labrador Sea or it is fully shifted towards high latitudes? Which role you expect GM to play in improving the NA SPG?

Page 13, line 17: Was the OHT computed through the meridional velocities or the atmospheric heat flux?

Section5: were the EOFs computed for global fields or only for shown areas?

Page 17, line 19 (Summary): naming is incorrect: control simulation is only 25 years

long and 2000 years simulation is attributed to as spin up throughout the text.

---

## Author Response (AR1)

**Response to reviewers comments on gmd-2020-43 "FORTE 2.0: a fast, parallel and flexible coupled climate model"**

We would like to thank both reviewers for their time and effort in reviewing our manuscript. Their suggestions have helped to clarify several points and improve the description of FORTE2.0 in many areas. Reviewer 1 requested a second simulation (without the stratosphere) to be presented alongside the original. This has resulted in numerous changes to the text and figures throughout the manuscript. Where we have added/changed figures we note that it is easier to read the captions in the revised submission than in the latexdiff-generated pdf.

The remainder of this document comprises a point-by-point response to the individual reviewers comments followed by a latexdiff-generated version of the manuscript which highlights all the changes made between the original submission and the revised version of the manuscript. The reviewers comments below are reproduced in black text and our responses are given in red. Where specific points have been addressed we identify the relevant change(s), referring to the page (P) and line numbers (L) in the latexdiff-generated pdf document accompanying this response.

**Response to RC1**

We thank the reviewer for their time and effort in reviewing our manuscript.

1/ As it is presented, FORTE2.0 is an intermediate resolution model like many others. The fact that the model is relatively fast does not appear clearly as an advantage compared to similar models that should be more or less as fast if they are run at the same resolution. If I understand well, the tool seems to have two main advantages. First, the model can include or not the stratosphere. However, the stratospheric dynamics is not at all discussed. The minimum for me would be to present simulations with and without resolved stratosphere and see the impact of this choice on model result. 2/ It seems that the model is readily configurable, providing maybe more flexibility than other tools. This interesting aspect is mentioned but not developed enough to see if this is a real strength of the tool. The manuscript should thus insist much more on those potential strengths (and maybe on others).

FORTE 2.0's performance, and the use of the term 'Fast' is in part to retain the connection to earlier studies which used FORTE, and in part due to the choice of the components, which when FORTE was originally coupled were indeed 'fast'. The IGCM (Forster et al., 2000, Joshi et al., 2015) is a spectral atmosphere, and spectral models are well-known to integrate much faster than grid-point models. At the time it was written, MOMA was shown to yield significantly faster performance on array processors than the GFDL MOM code upon which it was based (Webb 1995).

To avoid repeating the description of IGCM4 (Joshi et al 2015) we referred the reader to that publication for details of the stratospheric dynamics. We have expanded the description of the atmosphere and make it clear to the reader where they may find more information (P3 L17-19).

We have performed a second simulation using the L20 configuration, and have now integrated a comparison of the simulations with and without the stratosphere into the manuscript. This has resulted in many changes throughout the text and figures, all of which are highlighted in the accompanying latexdiff-generated pdf accompanying this response.

FORTE2.0 is readily configurable. We refer in our introduction to previous studies that have used FORTE in a variety of configurations to illustrate this point. It is difficult to quantify configurability, but we expand the text to give some more details on this element of FORTE2.0 (P2 L6-19).

2/ The model evaluation is very general. Many times it is mentioned that the results of the model are satisfactory or within the range of other models but the model performance itself is not really quantified. The observations are shown for some quantities for comparison with model results (some figures in section 4.2, figures 12-14) but not for many others (section 4.1, figure 14). This makes the evaluation harder to follow. Furthermore, it is not clear from the text if the simulation presented is from a 'standard configuration' of the model that may be used as a future reference or just an illustrative version that is not supposed to last and will not be used later. If the configuration presented is not a standard one, this strongly diminishes the interest of any diagnostic performed with this configuration and I would recommend that such a standard version is obtained before presenting it.

Hardware and compiler differences aside, the code and configuration of FORTE that has been archived will reproduce the (now two) simulation(s) presented in this manuscript (using the L20 and L35 configurations of IGCM4). In that sense, this could be considered a 'standard configuration', and by archiving the code/configuration and publishing the resulting simulations it may be used as a future reference. As we state in the manuscript, we have not exhaustively calibrated the model, and 'better' climates almost certainly could be achieved. Furthermore, the potential applications of FORTE2.0 are wide-ranging, and we expect that many users will define their own 'standard configuration' adapted to suit their purposes. When considering the plots for the manuscript we tried to strike a balance between the number of figures/subpanels and what we think the readers would be interested to see. In the revised manuscript we now present difference plots between the modelled SAT and 20CR (Compo et al., 2011) (new Figure 4) and between the modelled summer and winter precipitation and 20CR (new Figure 5). All data and scripts for plotting the figures in the manuscript are provided as supplementary material, so it is possible to reproduce and compute differences from other models and observational datasets.

Compo et al., "The Twentieth Century Reanalysis Project" QJRMS, 37, 654, Part A, 2011, Pages 1-28

3/ The model has no interactive sea ice. This is a strong limitation compared to similar tools and this should be mentioned earlier (in the abstract for instance) as this may be an important element for potential users.

The flux barrier that represents sea-ice is interactive. Once SST drops below a threshold the ice forms, its albedo changes, and it's surface temperature is calculated and updated. However, we do acknowledge that such a simple representation of sea ice may be a strong limitation for some users. We now state clearly that the ice is represented by a simple flux barrier in the abstract (P1 L3-4).

Specific comments

Page 1, lines 16- 21. The distinction between 'coarse resolution simplified models', 'intermediate resolution models' and the 'Earth Models of Intermediate Complexity (EMICs)' (introduced page 2, line 30) is to very clear to me. For instance, I would personally put ECBilt in the list of EMICs, and thus among the 'coarse resolution simplified models' rather than within the 'intermediate resolution models'.

We have moved ECBilt as you suggest (P1 L19).

Even if it is always better to use the most up-to date datasets, I do not think that using relatively old ones like the climatology of Levitus and Boyer (1998) - Levitus et al. (1998) (page 2, line 15) or the

heat transport of Trenberth and Caron (2001) (page 13, line 18) makes a big difference but stopping the analyses in 1997 for ENSO (page 16, line 11) may seem a bit too early to have a good evaluation.

We use the Levitus and Boyer (1998) and Levitus et al. (1998) T and S climatologies for the initial state of the simulation. Their use is historical, and as the reviewer suggests their age is unlikely to make a big difference to the result. For our ENSO analysis we now plot Gaussian distributions of the NINO 3.4 SST anomalies for both the L35 and L20 configurations, and for HadISST data extending from 1870-2019 (new Figure 13).

Page 3, line 28. It is not clear if the variable drag coefficient is applied both over land and ocean.

It is applied only over open ocean. We clarify this in the text (P4 L8).

Page 3, line 34. Not clear to me what is meant here by 'ice has melted' if there is no representation of sea ice

This has been rephrased to say "...will not reduce until the temperature rises above freezing point and the flux barrier deactivates." (P4 L13).

Page 4, line 9. What is the size of the 'polar island' ?

The polar island is a single row of ocean cells, so the northern extent of the ocean is 88°N. This detail is now added to the text (P4 L22).

Page 11, Figure 3. What is EN3? Not sure it is defined.

Thank you for spotting this. We had omitted the reference, and this is now given (P13 L9).

EN3 is a subsurface ocean T and S product from the UK Met Office which can be accessed here: https://www.metoffice.gov.uk/hadobs/en3/

Ingleby, B., and M. Huddleston, 2007: Quality control of ocean temperature and salinity profiles - historical and real-time data. Journal of Marine Systems, 65, 158-175 10.1016/j.jmarsys.2005.11.019

Page 13, Lines 9-14. Please specify where convection occurs in the model. The wording 'not uncharacteristic of coarse resolution ocean models' is another example of a general sentence where more substantial, quantified information would be required.

Convection occurs in MOMA where $d(rho)/dz < 0$. We do not routinely output where this takes place, but the mixed layer depth is an indication of where deep convection occurs. Winter mixed layer depths in the southern Labrador Sea reach 2500 m in a few grid cells. Winter mixed layer depths south of the Denmark Strait, Iceland and the Faroe Bank Channel can reach 1000 m. Wintertime convection is too shallow in the Nordic Seas, with mixed layer depths reaching 125-150 m in the central and eastern Nordic Seas. We have improved the text describing the regions of deep convection

(P16 L8-11) and added a new figure to show the winter mixed layer depths in both the L35 and L20 configurations (i.e. simulations with and without the stratosphere) (new Figure 11).

Page 16, line 11. The evaluation of ENSO characteristics is based on a figure from a paper published in 1997 while it would be very easy to evaluate precisely the simulated Nino3.4 index compared to observed one.

We now compare the distributions of the Nino 3.4 SST index from both FORTE 2.0 simulations and an up-to-date time series from the gridded observation-based product HadISST (new Figure 13). The distributions simulated by FORTE 2.0 are too narrow compared with observations, and extreme values are approximately 0.5C too small. We have updated the text which discusses the ENSO (P18 L27-32)

Page 17, line 6. It is mentioned that '. Comparison of the corresponding principal component time-series (Fig. 12) suggests the presence of some higher frequency variability in observations that is not captured by FORTE 2.0'. This should be quantified both for the AO and the NAO.

The suggestion of higher-frequency variability in the AO principle component time series is a visual artefact because of the longer time series of the 20$^{th}$ Century Reanalysis (140 years vs the 100 years of FORTE2.0 that we present). We removed the sentence from the manuscript.

**Response to RC2**

We thank Dmitry for his time and effort in reviewing our manuscript.

A general question before downloading the code: in which programming language is FORTE written?

FORTE is written in FORTRAN. This is an important point, and we have added a statement to the abstract to make this clear to the reader (P1 L4).

Page 3, lines 2 and 3: It sounds like two different grids are being used in the atmosphere. I would therefore rephrase to something like "A longitudinally regular and Gaussian in latitude grid with a grid spacing of _2.8_ is used for advection and diabatic processes."

This sentence has been reworded as suggested (P3 L12-13).

Page 3, line 31 and below: "FORTE 2.0 does not include dynamic sea-ice representation. Instead, sea ice is represented by a barrier: : :" Do I understand this correctly that there is no dynamic sea ice nor the sea ice itself but the flux barrier? I am curious about the process of flux computation in this place. How the values for temperatures and albedo to parameterize the presence of sea ice were chosen? At the end of the page the authors say "...until the ice has melted". Considering what is said above, is it

the same as ... until the atmospheric temperature becomes > 271_K? I assume the restoring of SST below the sea ice a part of the ocean component? Maybe it is worth mentioning this since this chapter describes the atmospheric component.

Yes. There is no dynamic sea ice in FORTE2.0 at the moment. In response to a comment by Reviewer 1 we added a statement to the abstract (P1 L3-4) to make this clear to the reader. The sea-ice representation is arguably the largest shortcoming in the current configuration, and top of the list of things we would like to address in the future. The albedo for sea ice in the real world typically ranges from 0.5 to 0.7. It is a tunable parameter in FORTE2.0. The simulations presented use a value of 0.6. The flux barrier becomes active once the sea surface temperature reaches 271K, and from this point IGCM4 computes the surface temperature in 'ice covered' regions. If the surface temperature continues to fall the albedo can increase further to mimic the effect of snow cover on sea ice. In reality, accumulation of snow cover on sea ice acts to increase the surface albedo to values as high as ~0.9. Given the size of the grid cells in FORTE2.0 we opted for setting an upper bound of 0.8.

Page 4, line 13: How the topography was interpolated? Was it smoothed in between or not? conservatively?

The topography used for the simulations present and supplied in the repository is a bilinear interpolation of the HadCM3 bathymetry (Gordon et al., 2000). Due to the coarse grid this was followed by a manual process to adjust the widths of narrow channels. Note that with the Arakawa 'B' grid a single grid point channel does not permit flow, but diffusion of tracer quantities will occur.

Page 4, lines 18 to 20: Considering the model "biases" which are shown below in the paper I wonder whether the geometrical scaling of GM could improve the solution? Was there a run made without GM? Does it improve the SPG in the NA?

We have not implemented or tested a geometric scaling of GM, or run FORTE2.0 without GM. These are interesting suggestions, and we would be interested to explore these in future.

Page 5, lines 2 to 3: in the Table 1 the background vertical diffusivity is defined as a constant value but it is stability dependent here. Which mixing scheme is used above the background?

This was an error in Table 1. We have removed the vertical diffusivity quoted in the table (P5 Table 1). In correcting this we noted that two of the other quantities (isopycnal tracer diffusivity and the steep slope horizontal diffusivity) in the table did not reflect the values used in the simulations presented. We have corrected these also.

The background vertical diffusion is stability dependent following Gargett (1984), as we described in paragraph on P5 L1-8.

Page 5, section 3: it is worth repeating that the pre-industrial atmospheric concentrations of CO2 were used.

This is now done in the revised manuscript (P5 L14).

Page 5, line 16: should there be the minus sign? The ocean warms initially but cools towards the end of the control run. Actually most of models simulate higher than observed ocean temperatures even under pre-industrial forcing (e.g. Griffies et al. 2011, doi:10.1175/2011JCLI3964.1; Lucarini and Ragone 2011).

We meant to say +/- 0.2 W m$^{-2}$. From around 600 years onwards the long-time mean is slightly negative, but here we intended to say that the even on short timescales the surface heat flux imbalance in a control simulation remains within +/- 0.2 W m$^{-2}$. This is now corrected (P5 L20).

Page 5, line 17: is the salinity trend caused by the use of the linear free surface (W_surf * SSS)?

This is a good suggestion. However, it is not the case. Taking into account the linear free surface the volume average salinity is of the order 1e-5 PSU higher after 2000 years. There is a small positive trend in the linear free surface, such that after 2000 years the global mean sea surface is 2.4e-7 m higher.

Page 6 line 8: the drop in AMOC happens abruptly. Do you have any idea of what has happened?

We believe the sudden drop is linked to the establishment of a fresh surface signature across the GIN and Irminger Seas (Figure 7f). The abrupt drop in AMOC has been a 'feature' of FORTE simulations, and it continues with FORTE2.0. As you note later, the common "cold bias" around Newfoundland is actually a warm anomaly in FORTE, and this extends into the Labrador Sea. This limits the dense water formation in FORTE2.0 to the Irminger and GIN seas. Hence the fresh anomaly that develops over these regions has a pronounced effect on the AMOC strength. We have added some text to explain this in the revised manuscript (P9 L 11-13).

Page 8 line 5: Is there a link between low ACC and GM (or is it because of winds)?

There is a link between GM and ACC strength. Kuhlbrodt et al (2012) examine coupled models submitted to IPCC AR4, and show that there is greater sensitivity of the ACC strength to the value of $k$ (GM thickness diffusion) than there is to the zonal wind stress. Our simulations use a value of $k$ = 2000. Looking at Fig. 1b of Kuhlbrodt et al (2012) FORTE2.0 has a stronger ACC than other models that use $k$ = 2000. It is closer in strength to the models using values of $k$ = 700-1000.

We have expanded the text discussing the ACC to include discussion of the link between GM and ACC strength (P9 L17-20).

T. Kuhlbrodt, R.S. Smith, Z. Wang, J.M. Gregory "The influence of eddy parameterizations on the transport of the Antarctic Circumpolar Current in coupled climate models" Ocean Modelling 52-53 (2012) 1–8

Page 8 line 9: I wonder why only 25 years? Do you expect any change in results if a longer period is considered?

Analysis of the climate state, either from models or observations is typically done for periods of a few decades. The choice of 25 years was arbitrary, and we would not expect any substantial change if a different duration or period was chosen.

Page 10: A pre-industrial run is compared with the present day climatology. Bias seems to be not the proper word then. Maybe it is also worth mentioning that the Levitus climatology is being used as a metric only.

I see your point about the use of the word 'bias' in this case. We have instead used the word 'anomaly' (P13 L10). Whilst addressing this point, we realized that 'Levitus climatology' should actually read 'EN3 climatology', as this is what is presented in Figure 7 (new Figure 8). We have rectified this.

Page 11, line 13: the commonly observed "cold bias" around Newfoundland is replaced by a warm anomaly instead. Something very different from most of the climate models is happening there. Same in SSH (page 13, line 5). Is it because of the winds? Does FORTE depict any MLD in the Labrador Sea or it is fully shifted towards high latitudes? Which role you expect GM to play in improving the NA SPG?

The extent of the subpolar gyre in this simulation is confined very much to the west. Correspondingly you can see the 10C and 14C contours in Fig. 7a extend further northwest than in EN3. The winds may play a factor in the structure seen here. FORTE2.0 does show some deep MLD in the Labrador Sea. We have monthly output for the final 25 years of the simulation, and have now added a figure of MLD for both the L35 and L20 configurations to the manuscript. In the period of monthly output we have, the L20 configuration shows stronger convection in the Labrador Sea and more generally across the high latitude N. Atlantic, whilst the L35 configuration depicts less extreme MLD that also occur further north into the Labrador Sea. As you alluded to earlier, a spatial/geometric scaling of GM would enable the user to choose GM that is more appropriate at both high and low latitudes, rather than seeking a compromise. This could improve the representation of the N. Atlantic subpolar gyre. Numerous studies, e.g. Decau and Myers (2005), Beismann and Redler (2003), have investigated the role of GM on N. Atlantic circulation and found positive results.

Deacu, D. and Myers, P.G. "Effect of a Variable Eddy Transfer Coefficient in an Eddy-Permitting Model of the Subpolar North Atlantic Ocean", Journal of Physical Oceanography (2005) 35 (3): 289-307.

Beismann, J-O. and Redler, R. "Model simulations of CFC uptake in the North Atlantic Deep Water: Effects of parameterizations and grid resolution", Journal of Geophysical Research, Vol. 108, C5, 3159

Page 13, line 17: Was the OHT computed through the meridional velocities or the atmospheric heat flux?

It was computed through the meridional velocities.

Section5: were the EOFs computed for global fields or only for shown areas?

We followed the method of Hurrell (1995) for the AO and NAO analysis. We now state in the revised text over which regions we compute our EOFs (P21 L2, P21 L11-12 and P22 L2).

Page 17, line 19 (Summary): naming is incorrect: control simulation is only 25 years long and 2000 years simulation is attributed to as spin up throughout the text.

Thank you, this is now corrected (P22 L7).

[revised manuscript text omitted]

---

## Author Response (AR2)

Dear Julia,

Thank you for your final comments on our manuscript, and for prompting us to carefully review the model description once more.

Forte 2.0 is constructed from IGCM4 and MOMA. You need to make it clear which versions of these models you are using. Both references for these models that are cited in the manuscript are a few years old (2015 for IGCM4 and 1996(!) for MOMA). Please update these so that your manuscript includes, or references, model description of changes in these models relevant to the versions you are using.

Despite the age of the references, the codes used are in fact little changed from those published. We have carefully worked through the model description (section 2) once more and can confirm that all the modifications to the code are now described.

Compared with the description of FORTE given in the technical document of Sinha and Smith (2002) the horizontal resolution has been increased by a factor of 2. We now state this (p3 lines 1-2).

We have introduced two new paragraphs, one detailing which fields are exchanged between the component models (p3 lines 4-9), and another that describes the coastal tiling routine (p4 lines 13-21).

We also clarified the description of the river catchment and runoff (p4 lines 5-7).

Finally, the version of MOMA we use is OpenMP parallelised, whilst the one described in Webb et al (1996) was not. This is now clarified (p5 lines 9-10).

All changes are highlighted in pages 3-5 of the latexdiff-derived version of the manuscript (appended below), and the page and line numbers above correspond to this version also.

I had thought that the original description paper for FORTE might help us out here, but I see that it is grey literature. Is it even relevant? Was FORTE constructed from these same 2 models?

Sinha and Smith (2002) is relevant, as much of the technical detail described applies equally to FORTE 2.0. We include a copy of this report in the Documentation folder that is now in the repository.

FORTE, as originally documented in Sinha and Smith (2002), used a very similar MOMA code to the present one in FORTE 2.0. All the pertinent changes since Sinha and Smith (2002) are documented in our section 2.2. The atmosphere code has changed from IGCM3 (Forster et al., 2000) to IGCM4 (Joshi et al., 2015). The key differences between the two versions are the radiation scheme and the IGCM4 code is parallelised.

I understood that you were going to include some of the background documentation, but I can only see the model code in the Zenodo archive.

I added the Documentation folder to the GitHub repository following the initial submission, but I wasn't aware until you queried it that I needed to create a new release in order for it to become visible through Zenodo. I've done this now (v2.0.1), and the Documentation is now visible in Zenodo.

The documentation will continue to evolve, particularly as we now have new users accessing and using the model. However, the model development is currently unfunded so we cannot prioritise this task.

Outside of FORTE 2.0, neither the ocean or atmosphere codes are subject to any formal version control. The Zenodo archive and this publication therefore represent the first step in implementing version control.

Yours sincerely,

Adam Blaker

[revised manuscript text omitted]
  changes being an increase in horizontal resolution of both the ocean and atmosphere components by a factor of 2, and an update of the atmosphere code from IGCM3 (Forster et al., 2000) to IGCM4 (Joshi et al., 2015).

The ocean and atmosphere components of FORTE 2.0 are coupled once per model day using OASIS version 2.3 (Terray et al., 1999) and PVM version 3.4.6 (Parallel Virtual Machine, see http://www.csm.ornl.gov/pvm/, Geist et al. (1994)). Daily average quantities of the variables that are to be exchanged are stored in arrays, and at the end of each model day these are passed to the coupler. MOMA provides daily mean values of sea surface temperature, zonal and meridional velocities, whilst IGCM4 provides solar and non-solar heat fluxes, net fresh water flux, and zonal and meridional surface wind stresses. Interpolation between the ocean and atmosphere grids is performed by the coupler using a pre-computed set of weights to ensure conservation.

Integration is relatively fast (~100 model years per wallclock day on a 28 core 2.4GHz Intel Broadwell CPU) and the model can be run on a desktop computer, making it ideal for experiments where more complex higher resolution models are resource limited. The retention of the full primitive equations for fluid flow in both atmosphere and ocean allows more realistic simulations than possible with Earth Models of Intermediate Complexity (EMICs). In addition, FORTE 2.0 is readily configurable, allowing experiments with realistic and idealized configurations of coastlines, orography, and ocean bottom topography.

**2.1 The atmosphere component**

The atmosphere component of FORTE 2.0 is IGCM4 (Joshi et al., 2015), run with a T42 spectral resolution. A longitudinally regular and Gaussian in latitude grid with a grid spacing of 2.8° is used for advection and diabatic processes. The resolution is sufficient to enable stable climate integrations without the need for flux adjustments. There are two pre-configured choices for the number of vertical levels: a troposphere only atmosphere represented by 20 $\sigma$ levels (L20) which extends to around 25 km altitude, or a 35 $\sigma$ level configuration (L35) which includes the stratosphere and extends to around 65 km altitude. To avoid issues with $2\Delta z$ oscillations under certain conditions the NIKOSRAD radiation scheme in IGCM3 (used previously in FORTE) was replaced with a modified version of the Morcrette radiation scheme (Zhong and Haigh, 1995). For further details of the IGCM4 and its performance we refer the reader to Joshi et al. (2015) and references therein. The model is run with 96 (L35) or 72 (L20) time steps per day. Orography is derived from the US Naval 1/6[th] degree resolution dataset. IGCM4 is MPI parallelised, and at this resolution integration on 16-32 cores achieves the best performance.

Atmospheric convection is dealt with via a Betts-Miller scheme (Betts and Miller, 1993). Low, medium and high layer cloud and convective clouds amount are represented, based on a critical relative humidity criterion (see the Appendix of Forster et al. (2000)). The formula which determines low-level cloud amount has an additional factor of 50% compared to that used by Forster et al. (2000) to correct a cold bias within the tropical ocean which led to unrealistic circulation in the Pacific. In addition to variation with solar zenith angle (and hence latitude), sea surface albedo is increased away from polar regions to compensate for the absence of aerosols which would otherwise scatter incoming solar radiation. Land grid boxes are assigned a vegetation index, one of 24 pre-defined vegetation types, which determine the albedo and roughness length.

Coupling to the dynamic ocean model requires some changes to the surface boundary layer. In order to conserve water it is necessary to account for soil moisture and implement river runoff. Soil moisture for each land grid box is represented as a bucket, or reservoir, 0.5 m in depth. Excess water, i.e. when the volume of water is greater than the volume of the bucket, is accumulated and added to the ocean as runoff at each coupling timestep. The land surface is divided into catchment basins

5 and the accumulated runoff is distributed between a list of  predetermined atmospheric grid cells that lie over the ocean and represent river mouths. The catchment areas and river discharge points are derived from Weaver et al. (1998) as explained in Sinha and Smith (2002). Runoff accumulated over Antarctica is distributed uniformly over the ocean south of 55°S, as a simplistic representation of iceberg calving and melting. Additionally, land snow cover is capped at a maximum thickness of 4 m. Excess snow over Antarctica and the Arctic region is treated separately as an additional runoff term that

10 represents iceberg melting and calving. As with the soil moisture, runoff from excess snow over Antarctica is distributed uniformly over the ocean south of 55°S. Excess snow melt over the Arctic is handled similarly, with a uniform distribution over the ocean north of 66°N.

A coastal tiling routine is implemented in order to handle the differences between the atmosphere and ocean grids. Grid cells in the ocean are either ocean or land, whilst atmospheric grid cells can be ocean, land, or partial (i.e. they extend over both

15 ocean and land cells on the ocean grid). Atmospheric grid cells that wholly overlie ocean are updated using the normal IGCM4 boundary layer scheme and the SST from the ocean model that is exchanged through the coupler. Similarly, cells that wholly overlie land are updated using the IGCM4 land surface scheme. For partial atmosphere grid cells two sets of boundary layer variables (e.g. latent heat flux) exist. One set are updated like any other land point using the IGCM4 boundary layer scheme, and the other set are updated like any other ocean point. The atmosphere then sees the weighted average of the heat fluxes

20 over the land and sea. Care is taken to ensure that atmospheric moisture is conserved and that precipitation is also apportioned correctly between the land surface scheme and the ocean fraction of the atmosphere cell.

[revised manuscript text omitted]